# A soft, self-sensing tensile valve for perceptive soft robots

Jun Kyu Choe [1,4], Junsoo Kim [2,4], Hyeonseo Song[1], Joonbum Bae [2] ✉ & Jiyun Kim [1,3] ✉

Soft inflatable robots are a promising paradigm for applications that benefit from their inherent safety and adaptability. However, for perception, complex connections of rigid electronics both in hardware and software remain the mainstay. Although recent efforts have created soft analogs of individual rigid components, the integration of sensing and control systems is challenging to achieve without compromising the complete softness, form factor, or capabilities. Here, we report a soft self-sensing tensile valve that integrates the functional capabilities of sensors and control valves to directly transform applied tensile strain into distinctive steady-state output pressure states using only a single, constant pressure source. By harnessing a unique mechanism, "helical pinching", we derive physical sharing of both sensing and control valve structures, achieving all-in-one integration in a compact form factor. We demonstrate programmability and applicability of our platform, illustrating a pathway towards fully soft, electronics-free, untethered, and autonomous robotic systems.

Soft robots that use intrinsically compliant materials have been in the spotlight for their ability to provide adaptability to unstructured environments and safety while interacting with humans[1]. Compared to their rigid counterparts, soft devices and actuators easily achieve complex motions[2,3] and simplify certain mechanical tasks, such as grasping[4], navigation[5–8], and rehabilitation[9,10]. The actuation of soft robots has been demonstrated by pneumatic[11–16] or hydraulic[17–23] pressure, tendons[24,25], or stimuli-responsive materials[2,26–28], nonetheless pneumatically actuated robots with inflatable elastomeric chambers have drawn considerable attention due to their simple fabrication and actuation[29].

Perception can enable soft robots to further leverage their mechanical compliance to achieve autonomous adaptation and effective interaction with constantly changing environments and humans[30,31]. However, to achieve this, soft pneumatic actuators have conventionally relied on hard solenoid valves and bulky electronic components[32], creating general hurdles for practical and potential

applications. Specifically, sensors, controllers, and regulators must be connected via complex codes and wires, yet still integrating rigid electronic components into soft bodies undermines the primary benefit of mechanical compliance and adaptability, whereas tethering soft robots with spatially isolated electronics limits their operating range. As a result, applications are greatly limited including human-robot interactions (HRIs), or in environments where electronic devices are difficult to use, such as in vivo, underwater, or in the presence of spark ignitions or high radiations. Therefore, these limitations emphasize the need for soft robots with well-integrated sensors, controllers, and valves in a soft and compact form.

As a step towards this goal, recent efforts have created soft analogs of individual components, including electronic skins and soft strain sensors capable of self-sensing for monitoring robot and human motions[10,33–39], and soft valves that can create nonmonotonous motions[40–51] and feedback control[4,47,50,52,53]. However, soft sensors alone still need other electronic components for control, while most state-of-

[1]Department of Materials Science and Engineering, Ulsan National Institute of Science and Technology (UNIST), Ulsan 44919, Republic of Korea. [2]Department of Mechanical Engineering, Ulsan National Institute of Science and Technology (UNIST), Ulsan 44919, Republic of Korea. [3]Center for Multidimensional Programmable Matter, Ulsan National Institute of Science and Technology, Ulsan 44919, South Korea. [4]These authors contributed equally: Jun Kyu Choe, Junsoo Kim. ✉e-mail: jbbae@unist.ac.kr; jiyunkim@unist.ac.kr

the-art soft valves fundamentally operate as on/off switches, which limits their perceptive capabilities to interact with continuously changing environments. Besides, physically combining single- function devices to create one multifunctional structure could additionally increase the structural complexity or form factor, which could limit the potential design space of perceptive soft robots.

Here, we developed a soft material-based self-sensing tensile valve (STV) capable of self-sensing and proportional control of soft pneumatic actuators from a single, constant supply pressure (Fig. 1a). The STV bridges sensing and control into one soft compact form with a programmable processor embedded in its structure, replacing the conventional requirement of complex assembly and connection of multiple devices. The STV is fabricated using a straightforward multistep assembly process (see Methods section 'STV fabrication' and Supplementary Fig. 1) comprising elastomeric inner and outer tubes, wrapping helical yarns (WHYs) that surround the inner tube, and 3D printed soft connectors (i.e., inlet-outlet and chamber connector) at each end (Fig. 1b(i)). The soft connectors determine the airflow directions inside the tubes by assigning the cavities between the inner and outer tubes as the inlet channel and the cavity of the inner tube as the outlet channel (Fig. 1b(ii)). This valve harnesses a unique mechanism, which we call "helical pinching", in which the WHYs elastically and reciprocally deform the channels as a continuous function of tensile strain. At zero strain $\varepsilon = 0$, WHYs helically wrap the

inner tube with a diameter equivalent to the inner tube diameter, and airflow through the inlet channel is proximately restricted, forcing the chamber pressure $P_{ch}$ to equalize with the surrounding atmospheric pressure $P_{atm}$. (Fig. 1b(i), (ii)). Upon application of an axial tensile strain, the WHYs straighten to deform the inner tube radially, which correspondingly increases/decreases the cross-sectional area and thus decreases/increases airflow resistance of the inlet/outlet channels, allowing intermediate flow through both inlet and outlet channels. When the STV reaches its maximum strain $\varepsilon = \varepsilon_{max}$, the airflow through the outlet channel is restricted, allowing $P_{ch}$ to reach the constant supply pressure $P_s$ (Fig. 1b(iii), (iv), and Supplementary Movie 1).

This single mechanism enables simultaneous and reciprocal transformations of both inlet and outlet channels, allowing the STV to be designed in a compact, seamlessly integrated linear form (diameter = 5 mm) advantageous for embodiment into soft systems and HRIs. A detailed comparison of the STV with state-of-the-art is shown in Supplementary Table 1. To further understand the underlying mechanism and aid future design, we rationalize structural and fluidic mechanics acting during tensile extension through simulations using finite element analysis (FEA) (see Methods section 'Finite element simulations' and Supplementary Movie 2), computational fluid dynamics (CFD) (see Methods section 'Computational fluid dynamics' and Supplementary Fig. 2), and analytical models. By leveraging these

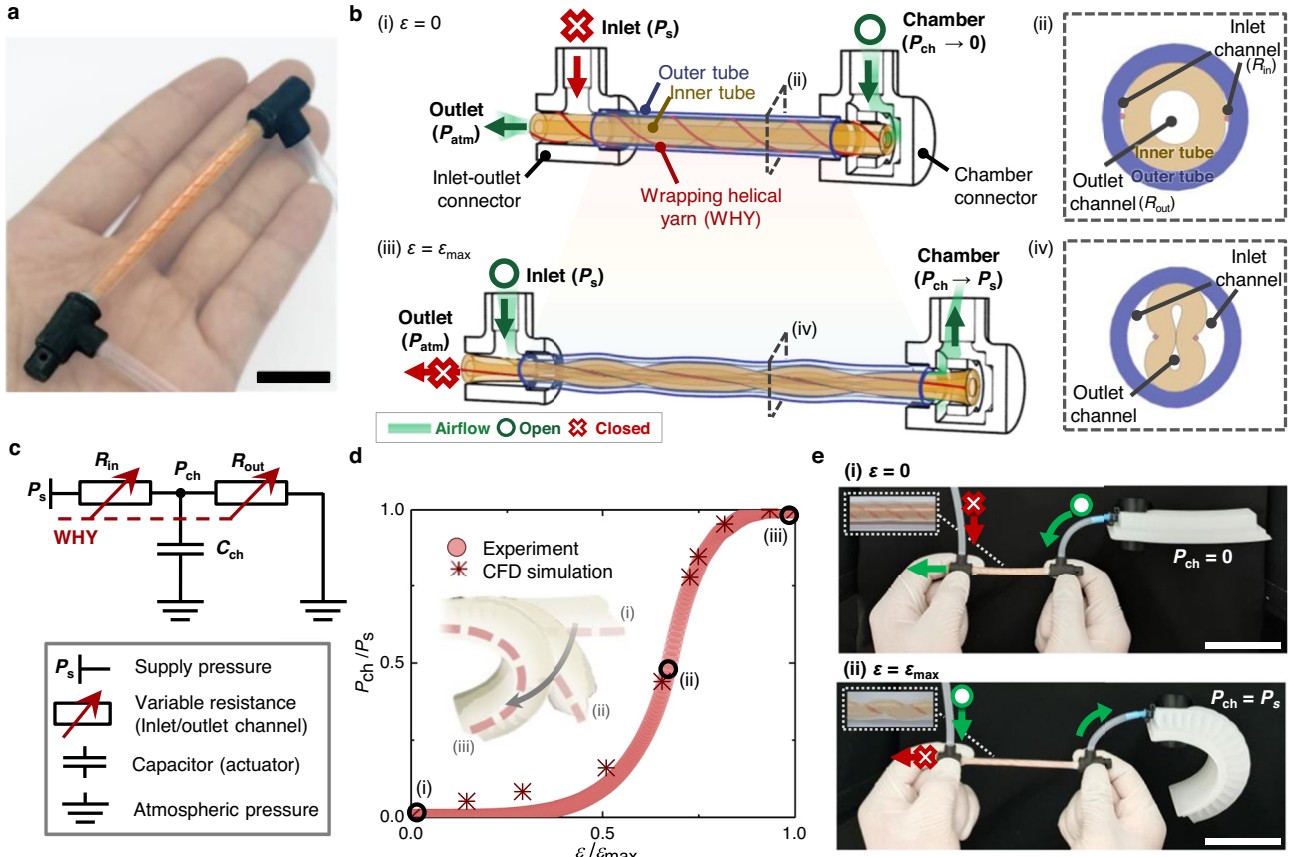

**Fig. 1 | Soft self-sensing tensile valve (STV) transducing strain into manageable proportional output pressures. a** Photograph of the fabricated STV. Scale bar = 3 cm. **b** Schematic diagrams of a projected STV and finite element analysis (FEA) cross-sectional results when tensile strain is not applied $\varepsilon = 0$ (i, ii) and is fully applied $\varepsilon = \varepsilon_{max}$ (iii, iv). The valve consists of an elastomeric inner tube that is tied with wrapping helical yarns (WHYs), an enclosing elastomeric outer tube, and connectors at the ends that regulate the path of the airflow. The air from a pressure supply flows inside the STV in the following directional path: inlet ($P_s$) - inlet channel

- chamber ($P_{ch}$) - outlet channel - outlet ($P_{atm}$). **c** An analogous electrical circuit that represents the STV pressure control mechanism. **d** Experimental and computational fluid dynamics (CFD) simulation results of the output chamber pressure to supply pressure ratio $P_{ch}/P_s$ as a function of the normalized strain $\varepsilon/\varepsilon_{max}$. Inset: schematic of a soft pneumatic actuator at $P_{ch}/P_s = 0$ (i), $P_{ch}/P_s = 0.5$ (ii), and $P_{ch}/P_s = 1$ (iii). **e** Operational photographs of the STV-controlled soft actuator when the tensile strain $\varepsilon = 0$ (i) and $\varepsilon = \varepsilon_{max}$ (ii). Insets: close-up images of the STV. Scale bars = 5 cm.

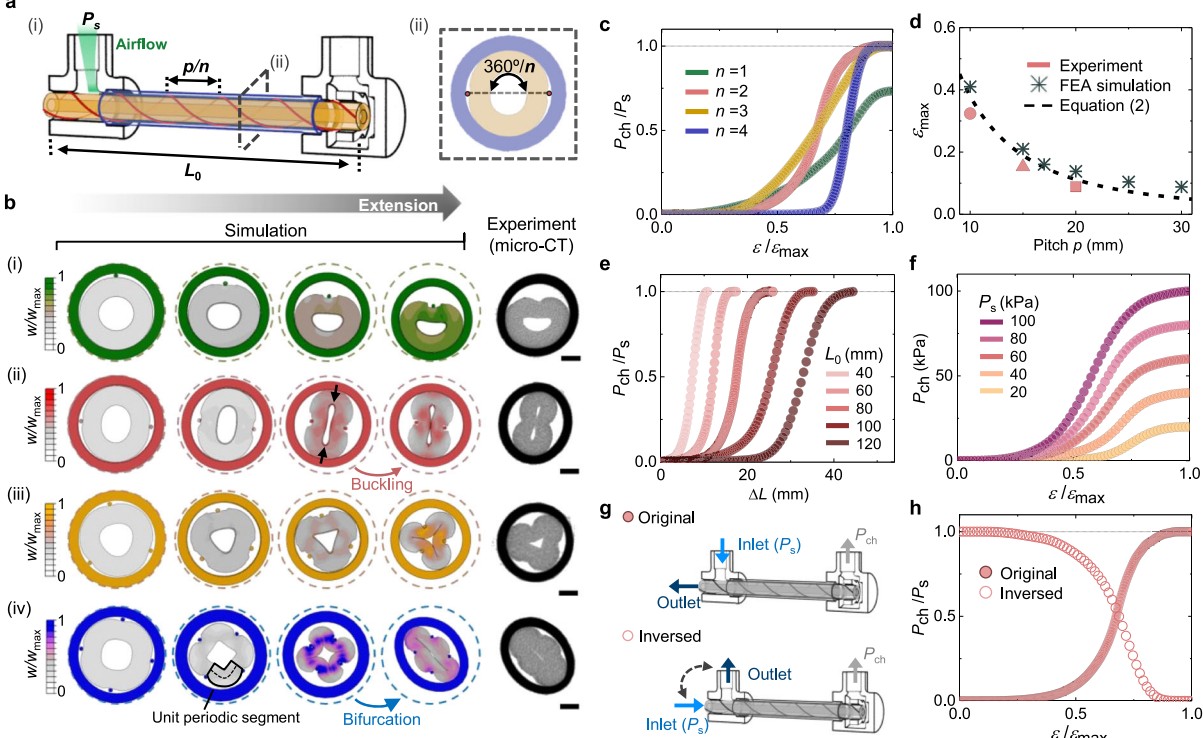

**Fig. 2 | Programming output chamber pressure curves. a** Schematic of the design parameters (i) and cross-sectional description (ii). **b** FEA cross-sectional results with contours of the normalized elastic strain energy density $W/W_{max}$ upon tensile strain and microcomputed tomography (micro-CT) images at $\varepsilon = \varepsilon_{max}$ for $n = 1$ (i), $n = 2$ (ii), $n = 3$ (iii), and $n = 4$ (iv). Scale bars = 1 mm. **c** Normalized chamber pressure $P_{ch}/P_s$ plotted against normalized strain $\varepsilon/\varepsilon_{max}$ for STVs with different numbers of WHYs $n$. **d** Maximum strain $\varepsilon_{max}$ as a function of the cyclic pitch of WHY $p$ ($n = 2$, $L_0 = 80$ mm). **e** Normalized chamber pressure curves as a function of extension length $\Delta L$ with different initial lengths traced by the center of WHY $L_0$ ($n = 2$, $p = 10$ mm). **f** Chamber pressure $P_{ch}$ plotted against normalized strain $\varepsilon/\varepsilon_{max}$ for different supply pressures $P_s$ ($n = 2$, $p = 10$ mm, $L_0 = 80$ mm). **g** Schematics of the STVs with original (top) and inversed (bottom) connections, and their normalized chamber pressure $P_{ch}/P_s$ are plotted against the normalized strain $\varepsilon/\varepsilon_{max}$ in (**h**) ($n = 2$, $p = 10$ mm, $L_0 = 80$ mm).

quantitative results, we show that various pressure curves for specific tasks can be programmed into the STV. Finally, we show two prototypical applications that demonstrate the potential of the STV to pave the way for untethered, electronics-free proportional applications: a soft gripper with a continuously adjustable holding force (Supplementary Movie 4) and a soft exosuit that autonomously adjusts the elbow assist torque according to the elbow angle (Supplementary Movie 5).

## Results

### Analog control of pneumatic actuation

The STV transforms the applied strain into distinctive steady-state output chamber pressure states. These chamber pressure states can be understood through an analogous electrical voltage divider (Fig. 1c) where the inlet and outlet channels of the STV each act as a variable resistor controlled by the helical pinching of WHY, the actuator connected to the chamber port acts as a pneumatic capacitor, and the connectors act as wires (the supply pressure and the atmospheric pressure are connected by the inlet-outlet connector, and the actuator chamber port and both channels are connected by the chamber connector). We derived an analytical expression for the chamber pressure $P_{ch}$ using the Hagen–Poiseuille equation as follows (see Methods section 'Chamber pressure'):

$$P_{ch} = P_s \frac{1}{R_{in}/R_{out} + 1}, \tag{1}$$

where $R_{in}$ and $R_{out}$ represent the airflow resistance of the inlet and outlet channels, respectively. Initially, $R_{in}$ is much greater than $R_{out}$

($R_{in}/R_{out} \approx \infty$), and air flows from the actuator (capacitor) to the atmosphere until discharge ($P_{ch} = 0$). As tensile strain $\varepsilon$ is gradually applied to the STV, $R_{in}$ and $R_{out}$ become comparable, and air flows into both the actuator and atmosphere ($0 < P_{ch} < P_s$). When maximum strain $\varepsilon_{max}$ is applied, $R_{in}$ becomes much smaller than $R_{out}$ ($R_{in}/R_{out} \approx 0$); thus, air flows solely into the actuator until fully charged ($P_{ch} = P_s$).

Figure 1d shows the experimental data and CFD simulation results (see Methods section 'Computational fluid dynamics' and Supplementary Fig. 2) of $P_{ch}$ as a function of normalized strain $\varepsilon/\varepsilon_{max}$, further illustrating that $P_{ch}$ is a smooth, one-to-one correspondence curve suited for analog control. Therefore, through simple connection and stretching, the STV provides facile analog control of a soft pneumatic actuator (Fig. 1e and Supplementary Movie 1). The experimental data obtained at steady-state conditions (see Methods section 'soft tensile valve characterization') and CFD simulation show good agreement, demonstrating that our model is accurate.

### Pressure curve profile programming

The profiles of the pressure curves generated by our STV can be programmed within its geometrical design and further tailored by adjusting the setup conditions postfabrication. First, we investigated and analyzed the dependency on the geometric parameters: the number of WHY $n$, the cyclic pitch of WHY $p$ (mm), and the initial length traced by the center of WHY $L_0$ (mm) (Fig. 2a(i)). For $n \geq 2$, the WHYs are evenly distributed radially so that the angle between each WHY is $360°/n$ (Fig. 2a(ii)). To predict the deformation behaviors and further excavate the deformation mechanisms of STVs with various geometric parameters, we performed simulations via the FEA. Figure 2b shows FEA cross-sectional deformation results from $n = 1$ to

$n = 4$ with the contours of the normalized elastic strain energy density $W/W_{max}$ (see Methods section 'Finite element simulations') and characterization results by microcomputed tomography (micro-CT) (see Methods section 'Microcomputed tomography'). For the transversal view of the FEA results, see Supplementary Fig. 3. Upon tensile strain, the WHYs are gradually straightened to be aligned on the central axis of the undeformed inner tube until mechanically confined by the inner tube. This straightening motion pinches the inner tube, creating the helical structure of the inlet and outlet channels. These channels show distinct deformation behaviors depending on $n$, reducing $R_{in}/R_{out}$ in Eq. (1) in different routes to create unique $P_{ch}$ curve profiles.

We categorize these deformation behaviors as either deflection-dominated or buckling-dominated. For $n = 1$ and 3, the deformation is predominantly a gradual deflection of the channels, producing a gradual chamber pressure curve under tensile strain. Figure 2c and Supplementary Fig. 4a show these curves in normalized form ($P_{ch}/P_s$) as a function of the normalized axial tensile strain ($\varepsilon/\varepsilon_{max}$) applied to the STV. In these deflection-dominated deformations, the pressure curves with relatively lower slopes compared to those when $n = 2$ and 4 (Fig. 2c) are rationalized by the strain energy of the inner tube, increasing constantly from zero to a maximum without encountering energy barriers, as shown in Fig. 2b(i), (iii) and Supplementary Fig. 3a. We note that when $n = 1$, the outlet channel cannot be fully tightened due to the radial asymmetry so that $P_{ch}$ reaches $P_s$ (Fig. 2b(i)), resulting in a pressure curve with the lowest slope.

For $n = 2$ and 4, the deformation proceeds through buckling-induced sudden deformations of the channels, resulting in pressure curves with a relatively abrupt slope (Fig. 2c). In these buckling-dominated deformations, the strain energy of the inner tube increases from zero to a maximum and then decreases to attain a local minimum in the second stable state (Supplementary Fig. 4b). The instability when $n = 2$ mainly results from the interaction between the bending and compressive energies of the elastomeric tube as it deforms; that is, at the onset of buckling, the compressive forces shown as arrows in Fig. 2b(ii) switch from impeding bending of the inner tube to reinforcing it[54]. On the other hand, the instability when $n = 4$ results from the high energy barrier generated during when the WHYs bend unit periodic segments of the inner tube, as shown in Fig. 2b(iv). Since the neutral line length of the unit periodic segments decreases as the number of WHY $n$ increases, the bending of the segments requires greater energy[55]. Thus, for $n = 4$, the structure chooses bifurcation instead, rapidly switching the deformation shape similar to that when $n = 2$ to relax the local strain energy $w$, as shown in Fig. 2b(iv). This macroscopic buckling when $n = 4$ relaxes much higher normalized strain energy than when $n = 2$ (Supplementary Fig. 4b), producing the highest-slope pressure curve similar to an on/off switch where the chamber pressure $P_{ch}$ is either 0 or $P_s$[4,50]. We note that in terms of sensitivity, defined as $S = \delta(P_{ch}/P_s)/\delta\varepsilon$, a 14-fold tunability is achieved ranging from 3.34 when $n = 1$ to 40.66 when $n = 4$ (see Supplementary Fig. 4c and Methods section 'soft tensile valve characterization').

To validate our simulation results, we derived an analytical expression of the deformation contours for different $n$ values using large deformation beam theory (see Supplementary Note 1) and experimentally characterized the STV deformation behaviors using micro-CT. We found the simulation to be in reasonable agreement with the analytical model shown in Supplementary Fig. 8 ($n = 2$, 3, and $n = 4$ until bifurcation) and the cross-sectional micro-CT images shown in Fig. 2b.

Then, to account for the extension range of the STV in the design space, we considered the maximum strain $\varepsilon_{max}$ of the STV where $P_{ch}$ is saturated. Since the helical pinching mechanism of the STV relies on the straightening of the WHYs to deform the airflow channels, the maximum strain $\varepsilon_{max}$ is highly dependent on the initial straightness of

the WHYs, which can be evaluated in terms of the cyclic pitch of WHY $p$, given the value of the initial diameter of the inner tube. Assuming that the WHYs do not intrinsically stretch under tensile strain and that the inner tube cannot be stretched locally at both ends due to anchoring on the connectors, the maximum strain $\varepsilon_{max}$ for each number of WHY $n$ can be derived in terms of pitch $p$ using the Pythagorean theorem (see Supplementary Note 2 and Supplementary Fig. 9). For example, the analytical expression for $n = 2$ yields:

$$\varepsilon_{max} = \frac{L_0 - L_t}{L_0}\sqrt{\left(\frac{2\pi D_0}{p}\right)^2 - 2(r_0 + t_0) + 1} - 1, \quad (2)$$

where $L_t$ is the axial transition length of the inner tube accounting for the fixed boundary condition with zero strain at both ends of the inner tube, $r_0$ is the radius of the WHY, and $t_0$ is the thickness of the inner tube. The maximum strain $\varepsilon_{max}$ is calculated to be inversely correlated to the cyclic pitch of WHY $p$ (regardless of $n$). In Fig. 2d, we plotted Eq. (2) along with our FEA and experimental results and found good agreement. Finally, in Fig. 2e, we show that programming of the maximum strain $\varepsilon_{max}$ can be further leveraged to design the range of the extension length $\Delta L = L_0\varepsilon$ of the STV by tuning the initial length $L_0$. Since the change in pitch $p$ or initial length $L_0$ does not disrupt the cross-sectional deformation route of the inner tube in Fig. 2b, the maximum strain $\varepsilon_{max}$ and the extension length $\Delta L$ can be independently programmed without changing the shape of the pressure curve (see Supplementary Fig. 4d, f).

We note that the significance of the WHY is lost when stretching of the STV cannot reconfigure the WHY enough to generate radial forces to pinch the channels. As above, we found the design space of the STV from the experimental and simulation results to be $10\,mm \le p \le 30\,mm$ and $L_0 \ge 40\,mm$, as the material failed before generating significant deformation of the channels outside this range.

Next, we show that the chamber pressure curve can be further programmed by changing the STV setup conditions postfabrication. For instance, by changing the constant supply pressure $P_s$ to different levels, the maximum chamber pressure $P_{ch}$ of the STV is adjusted accordingly at $\varepsilon = \varepsilon_{max}$ (Fig. 2f). Furthermore, by inversing the inlet and outlet of the STV, as shown in Fig. 2g, the airflow direction passing through the channels is reversed. Therefore, the inlet and outlet channels are switched (see Supplementary Fig. 10 and Supplementary Movie 3) so that $R_{in}$ is now initially much smaller than $R_{out}$ ($R_{in}/R_{out} \approx 0$) but becomes much larger than $R_{out}$ ($R_{in}/R_{out} \approx \infty$) at $\varepsilon = \varepsilon_{max}$. Thus, from Eq. (1), we can achieve inverted pressure curves for every design of the STV, as shown for example in Fig. 2h.

## Untethered and electronics-free soft gripper

Now that we have modeled the design principles for the chamber pressure curves of our STV, we demonstrate the capability of the STV to provide precise and programmable control of soft actuators that can be integrated into untethered and electronics-free systems (Fig. 3a) using only a single, constant supply pressure $P_s$.

We designed a soft gripper with a continuously adjustable holding force using a $CO_2$ canister, a mechanical pressure regulator, an STV, a 3D printed handle, and two soft actuators (Fig. 3b). First, the high pressure emitted from the $CO_2$ canister was regulated to provide the STV with a constant $P_s$ of 60 kPa. Then, each connector of the STV was fixed to the body and the handle so that the STV could be tensioned by manually pulling the handle (Fig. 3c). Finally, the chamber pressure $P_{ch}$ coming out from the STV was transferred to two soft actuators to provide an adaptable holding force depending on the tensile strain of the STV.

Figure 3d shows that our soft gripper can create a holding force curve that can be continuously controlled up to 14.8 N at a constant supply pressure of $P_s = 60\,kPa$. In addition, by changing the constant supply pressure $P_s$ through simple readjustment of the mechanical

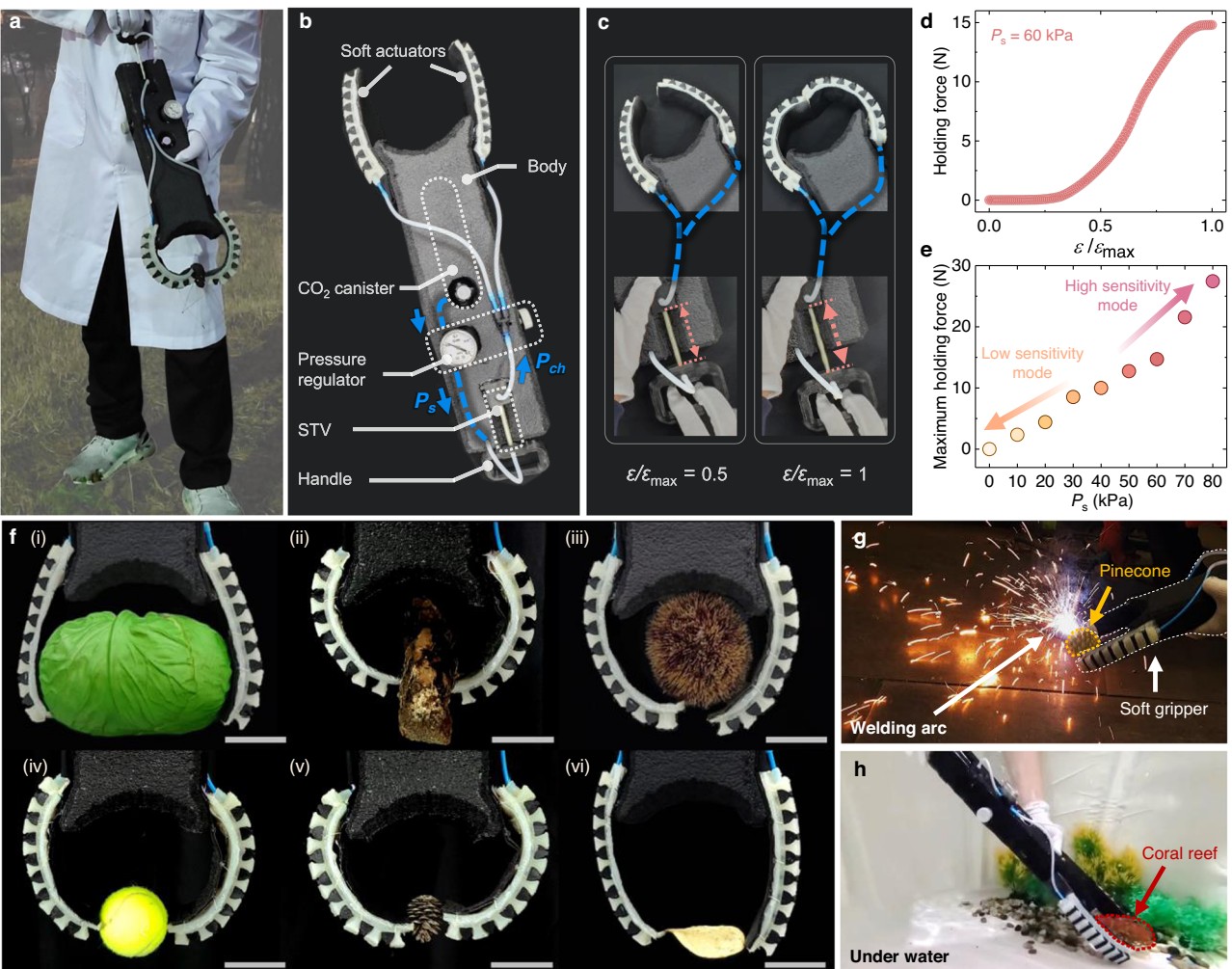

**Fig. 3 | Untethered and electronics-free soft gripper. a** A photograph of the soft gripper picking up a pinecone outdoors. **b** Details of the soft gripper. **c** Photographs of the soft gripper at normalized strain $\varepsilon/\varepsilon_{max} = 0.5$ (left) and $\varepsilon/\varepsilon_{max} = 1$ (right). The curvature of the soft actuators is continuously controlled by extending the STV ($n = 2$, $p = 10$ mm, $L_0 = 80$ mm) with the handle. **d** Holding force of the gripper plotted against the normalized strain $\varepsilon/\varepsilon_{max}$ at supply pressure

$P_s = 60$ kPa. **e** Maximum holding force of the gripper plotted against the supply pressure $P_s$. **f** Gripping demonstrations at $P_s = 60$ kPa: a cabbage (i), a tree bark (ii), a chestnut bur (iii), a tennis ball (iv), a pinecone (v), and a potato chip (vi). Scale bars = 5 cm. **g** Photographs of the soft gripper handling a pinecone near a welding arc (which can affect electronic circuits by electromagnetic interference), and (**h**) coral reef under water.

regulator, we show in Fig. 3e that different modes of our soft gripper can be programmed between two extremes with high maximum holding force with high sensitivity or high motional resolution with low sensitivity using the same STV (thus, independently of the STV pulling force). In Fig. 3f, we demonstrate the capability of our soft gripper at a constant supply pressure $P_s = 60$ kPa, readily simplifying a wide range of gripping tasks from lifting a cabbage that requires a strong holding force to lifting a potato chip that requires highly precise control to avoid breaking while gripping. Last, we show that integrating our STV into the body of a soft machine allows controllable operation of the gripper in untethered and electronics-free form, which can be useful in environments with spark ignition (Fig. 3g) or in underwater applications (Fig. 3h). We note that the STV was designed to provide continuous control with intermediate sensitivity $S$ (with $n = 2$), to be well fitted to the soft gripper body ($L_0 = 80$ mm) with high maximum strain $\varepsilon_{max}$ (with $p = 10$), and to increase gripper pressure according to strain (with original connection). However, by programming the STV with different design or setup conditions, we may achieve binary on/off control (with $n = 4$), a gripper with different strain sensitivity (by tuning $\varepsilon_{max}$), or a reverse action gripper (with inverse STV connection) for other specific tasks.

From a practical point of view, the targeting of the output chamber pressure was immediately set according to the strain of the STV, but for soft actuators, the achievement of the target pressure was strain rate-dependent that highly depended on the soft actuator characteristics such as the initial chamber volume (Supplementary Fig. 6a). The longevity of the $CO_2$ canister (95 g) using the STV was 64 min on average but could be increased to 100 min using an additional inlet resistor (Supplementary Fig. 6b). The STV showed good reproducibility with similar output from five different samples (Supplementary Fig. 6c), durability up to 10,000 full cycles, and could be simply repaired after sharp cuts or detachment of the outer tube (Supplementary Fig. 6d). See Supplementary Note 3 for more information related to the STV during practical use.

**Autonomous and self-adaptive soft exosuit**

The STV with a soft, compact, and linear form can also provide conformable and safe human–machine interactions (see Supplementary Fig. 7 for STV behavior under bending, buckling, kinking, and compression). To take advantage of this, we created a soft wearable robot that can sense human body movements and simultaneously process them for autonomous control. In Fig. 4a(i), a soft elbow exosuit that

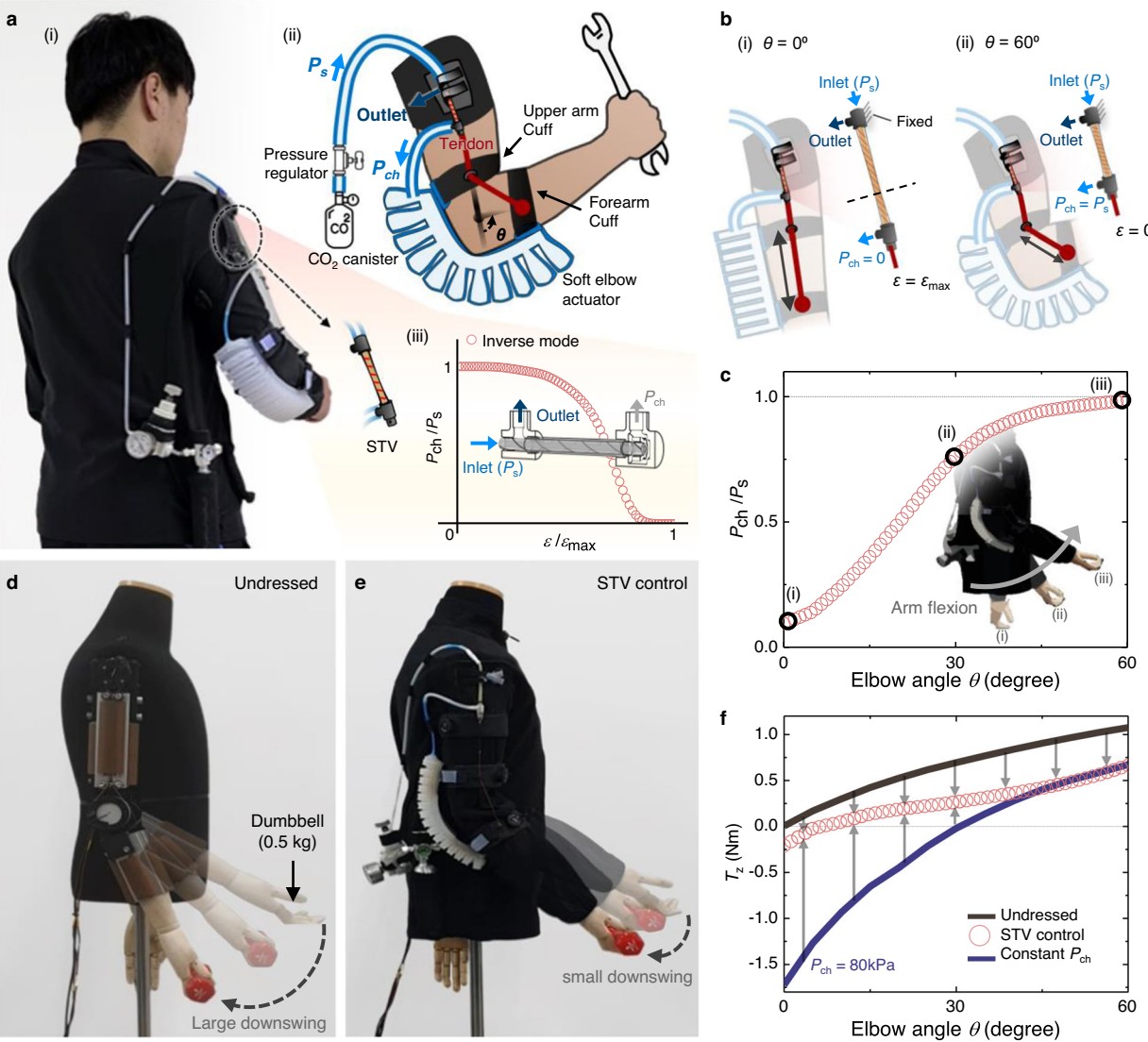

**Fig. 4 | Autonomous and self-adaptive soft exosuit. a** Overview of the soft exosuit. A photograph of the soft exosuit on a user (i) and details of the soft exosuit (ii). The STV ($n = 2$, $p = 10$ mm, $L_0 = 80$ mm) is connected in inverse mode, where the produced normalized chamber pressure $P_{ch}/P_s$ is a decreasing function of the normalized strain $\varepsilon/\varepsilon_{max}$ (iii). **b** Schematics of the exosuit and the STV at the start and end elbow angles. The STV reaches the maximum strain $\varepsilon = \varepsilon_{max}$ at $\theta = 0°$ (i) and releases strain $\varepsilon = 0$ to its original length at $\theta = 60°$ (ii). **c** Normalized chamber pressure $P_{ch}/P_s$ plotted against the elbow angle $\theta$. Insets: schematic describing the elbow angle when $\theta = 0°$ (i), $\theta = 30°$ (ii), and $\theta = 60°$ (iii). **d** Sequential photographs of a mannequin with an applied elbow torque of 1 Nm by a motor and subsequent gripping of a dumbbell (0.5 kg). **e** Results with the mannequin with our soft exosuit from the same procedures in **d**. **f** Assistive torque $T_z$ plotted against elbow angle $\theta$. The constant $P_{ch}$ (blue line) represents data with our soft exosuit, but without STV control (constant $P_{ch} = 80$ kPa).

adjusts the assistive torque according to the angle of elbow flexion is demonstrated to reduce the muscle load due to gravity.

The exosuit consists of a $CO_2$ canister, a mechanical pressure regulator, and an STV integrated into a resilient garment; a soft elbow actuator integrated into tightening cuffs (i.e., upper arm cuff, forearm cuff); and a tendon (Fig. 4a(ii)). The flexion of the elbow is detected as strain and transferred to the STV by the tendon. Similarly to the soft gripper demonstration, the STV with the parameters $n = 2$, $p = 10$ mm, and $L_0 = 80$ mm was used, yet we additionally took into account the restoring force of the STV under extension (see Supplementary Fig. 4i) for the soft exosuit, which can hinder the effective assistive torque generated by the soft elbow actuator. Specifically, we first connected the STV in inverse mode (Fig. 4a(iii)). Then, we tied the tendon to the chamber connector of the STV and the forearm cuff, passing through a routing ring in the upper arm cuff, as shown in Fig. 4b, so that the STV was fully strained ($\varepsilon = \varepsilon_{max}$) at (i) $\theta = 0°$ and restored ($\varepsilon = 0$) at (ii) $\theta = 60°$ as the distance between the cuffs decreased. In this way, we

achieved chamber pressure as an increasing function of the elbow flexion angle $\theta$, as shown in Fig. 4c, without incorporating the restoring force of the STV at high angles (see Supplementary Fig. 13).

To characterize our soft exosuit, we used a mannequin with a 1.6 kg forearm (an approximately average weight for adult males) and connected a motor and a torque sensor to the elbow joint of the arm. A torque of 1 Nm was applied to the motor to raise the elbow to $\theta = 60°$, and then a dumbbell (0.5 kg) was held by both the undressed mannequin (Fig. 4d) and the exosuit-dressed mannequin (Fig. 4e) for comparison. As a result, the elbow was less weighted down by the dumbbell with the exosuit, demonstrating the effective gravity assistance of our exosuit at high flexion angles.

To further characterize our exosuit in general, we measured the elbow torques required to lift the forearm over the useful range of motion of the elbow. In Fig. 4f, we show that the exosuit with STV control reduced the average torque by 63% compared to the undressed condition with only gravitational torque acting on the elbow (see

Methods section 'Soft exosuit'). In addition, applying the soft elbow actuator with a sole constant pressure $P_{ch} = 80$ kPa produced a high torque in the opposite direction to the gravitational torque at lower flexion angles. These results demonstrate that our STV with proportional control autonomously and effectively compensates for the gravitational torque without involving undesirable torque assistance. We note that the range of the extension distance $\Delta L = L_0 \varepsilon$ of the STV can be tuned, as shown in Fig. 2e, to account for different flexion angle ranges or to support other body parts.

## Discussion

By harnessing the helical pinching mechanism, a material-based approach to control pressure gradients in fluidic circuits, we demonstrated the STV that simultaneously self-senses its strain and generates corresponding output chamber pressures from a single, constant supply pressure. Helical routing structures have been previously explored to create various stretchable devices, including soft sensors and artificial muscles (see Supplementary Table 2), due to their stretchable nature as the fibers can be straightened out under tensile strain. Here, we exploited helical routing of yarns over a soft tube to develop the pinching mechanism that can control pneumatic resistance of the tube channel under tensile strain in a continuous and programmable manner. This mechanism is well identified by our computational and analytical models, which can further support the design of the output pressure curves produced by the STV for optimization in specific applications. We leveraged this knowledge to realize analog control of soft pneumatic actuators and develop a soft gripper with tunable holding force-strain sensitivity, and an exosuit that effectively and autonomously self-adapts its assistive torque. Using only low-cost (see Supplementary Table 3), widely accessible all-soft materials and a simple fabrication process, the STV facilely replaces the former requirement of electronic infrastructures for perceptive robots, opening the avenue for proportional applications of soft machines including in environments where electronics may not be compatible, such as in the presence of spark ignition, dust, or radiation, as well as in vivo or underwater environments. Furthermore, the STV mechanically integrates the sensing and subsequent feedback control in a compact, seamless form owing to the helical pinching mechanism. Therefore, the STV is anticipated to be more advantageous for complex actuation systems, such as multi-degree-of-freedom exosuit that simultaneously assists multiple body parts. Overall, this platform represents a step towards fully soft, electronics-free, untethered, autonomous, and perceptive robotic systems. Besides, our facile approach to implement perception for self-adaption to continuously changing environments may have further implications in material-embedded intelligent systems, such as tensegrities[25], structured fabrics[56], navigating robots[6], and mechanical metamaterials[57].

## Methods
### Fabrication of the samples
Each STV was fabricated using an elastomeric inner tube (GSJ) with an inner diameter of 2 mm and outer diameter of 4 mm, a semi-transparent silicone outer tube (Inner MED) with an inner diameter of 4 mm and outer diameter of 5 mm, a polyethylene 8-ply yarn (Mark), and 3D printed inlet-outlet and chamber connectors (J750; Stratasys) made from a mixture of soft materials—Tango and Agilus 30 (all Stratasys) in a 6:4 weight ratio—with the PolyJet technique.

First, to designate the location of the inner tube where the yarn will be helically wrapped, we used 3D printed guide stamps (Neo 800; Stratasys) fabricated with Somos Perform (Stratasys) to mark the inner tube with a fast-drying ink. After drying the ink for 5 min, we wrapped the 8-ply yarn following the guideline and fixed both ends with knots. We prepared a prepolymer mixture of ClearFlex 30 (Smooth-On) by mixing two components, namely, part "A" and part "B", in a 1:1 volume

ratio in a centrifugal mixer (Thinky) for 40 s at 400 G. Then, to stably fix the yarn to the inner tube, we dip-coated the inner tube in the prepolymer mixture and kept the tube in a vertical position at 60 °C for 2 days to cure the prepolymer. We note that the use of ClearFlex 30 resulted in an optimal coating thickness (50 μm) and uniformity (see Supplementary Table 4). After curing, the STV was prepared by assembling and gluing the other components in three consecutive steps: (i) connecting the inner tube into the chamber connector, (ii) inserting the outer tube, and (iii) connecting the inlet-outlet connector. In all steps, a flexible cyanoacrylate adhesive Permabond 2050 (Permabond) was used for gluing. More detail and a step-by-step illustration of the fabrication are shown in Supplementary Fig. 1.

### Chamber pressure
To characterize the dependence of the chamber pressure $P_{ch}$ of the STV on the supply pressure $P_s$ and the airflow resistance of the inlet channel $R_{in}$ and the outlet channel $R_{out}$, we first calculated the air velocity $v$ using $Q = (1/4)\pi v d^2$ where $Q$ is airflow rate obtained from data, and $d$ is the inner diameter of the inner tube. Then, the Mach number $M$ was obtained using $M = v/c$, where $c$ is the speed of sound. Reynold's number $Re$ was obtained using $Re = \rho v d/\mu$ where $\rho$ is the density of air, and $\mu$ is the dynamic viscosity of air. We estimated incompressible laminar flow with $M \approx 0.028$ and $Re < 1223$, which is smaller than the Mach number limit ($M \approx 0.3$) for compressibility[58] and the critical Reynolds number $Re \approx 2300$ for the transition to turbulence[59]. Therefore, the expression relating the flow rate of the inlet channel $Q_{in}$ and the outlet channel $Q_{out}$ to the pressure gradient and airflow resistance can be obtained using the Hagen–Poiseuille equation yielding:

$$Q_{in} = \frac{P_s - P_{ch}}{R_{in}}, \tag{3}$$

$$Q_{out} = \frac{P_{ch} - P_{atm}}{R_{out}}. \tag{4}$$

By solving these equations at steady state $Q_{in} = Q_{out}$ and setting $P_{atm}$ as the reference pressure ($P_{atm} = 0$), we obtain the expression for the chamber pressure given by Eq. (1), which allows us to predict the chamber pressure $P_{ch}$ to be in the range from 0 to $P_s$ depending on the ratio between $R_{in}$ and $R_{out}$.

### Soft tensile valve characterization
To measure the chamber pressure as a function of extension, we built a custom measuring instrument with a force/torque sensor (RFT64-SB01; Robotous), a linear ball guide (LX-15; Misumi), a motor (MX-106; Robotis), and 3D printed STV holders (Ultimaker). Constant pressure was supplied from a pneumatic system (VPPM; Festo), and the chamber pressure was measured with a pressure sensor (PSS-01 V-R1/8; Autonics). The overall assembly is shown in Supplementary Fig. 12. The connectors of the STV were fixed to each 3D printed STV holder, and then the STV was extended 0.1 mm at a time by the linear actuator. At each extension level, the measurements were averaged over 0.4 s after the steady state was reached. We repeated this process five times and then averaged the results. The sensitivity $S$ $\delta(P_{ch}/P_s)/\delta\varepsilon$ was calculated at the intermediate of the maximum $P_{ch}/P_s$ ($P_{ch}/P_s = 0.37$ for $n = 1$ and $P_{ch}/P_s = 0.5$ for $n = 2$–4).

### Microcomputed tomography
An X-ray micro-CT system (SkyScan 1176), a noninvasive CT scanner originally developed for the inspection of small animals, was used to obtain cross-sectional images of the STV. The STV samples were stretched and fixed with a jig and then placed on the micro-CT stage. CT images were collected at an X-ray voltage of 50 kV, a current of 500 μA, and an image resolution of 12.33 μm. Then, the CT data were

reformatted into cross-sectional images using a 3D reconstruction program (Osy 11).

## Finite element simulations

To predict the morphing behavior and strain energy density of the STVs under extension, FEA was performed using commercially available software ABAQUS.

First, to obtain the elastic property coefficients, we performed uniaxial tension tests (AGX-100NX; SHIMADZU) of the WHY, inner tube, and outer tube at a strain rate of 0.0006 s$^{-1}$. The inner and outer tubes were cut into dog-bone shapes for the test. The nominal stress-strain data of the WHY is shown in Supplementary Fig. 11a, and the nominal stress-strain data of the inner and outer tubes are shown in Supplementary Fig. 11b. For the WHY, the result was linearly fitted to obtain the Young's modulus of $E = 15.54$ GPa. For the inner and outer tubes, we fitted the experimental data into various hyperelastic models. For each model, we checked the Drucker's stability criterion defined by the following:

$$d\sigma : d\varepsilon \geq 0, \tag{5}$$

where $d\sigma$ is the change in stress due to an infinitesimal change in strain $d\varepsilon$. The Ogden and reduced polynomial models with specific term numbers ($N$) were found to meet the stability criterion and were selected as candidate models (Supplementary Fig. 11c, d). Their strain energy functions are as follows:

$$W_{\text{Ogden}} = \sum_{i=1}^{N} \frac{\mu_i}{\alpha_i} \left( \lambda_1^{\alpha_i} + \lambda_2^{\alpha_i} + \lambda_3^{\alpha_i} - 3 \right), \tag{6}$$

$$W_{\text{r\_polynomial}} = \sum_{i=1}^{N} C_{i0} \left( \lambda_1^2 + \lambda_2^2 + \lambda_3^2 - 3 \right)^i, \tag{7}$$

where $\lambda_n$ are the principal stretches, and $\mu_i$, $\alpha_i$, and $C_{i0}$ are material-dependent fitting parameters. The optimal fits for the operational strain range of the STV ($\varepsilon < 1$) were selected as the following models and parameter values: Ogden $N = 2$, with $\mu_1 = 2.989 \times 10^{-3}$, $\mu_2 = 1.734$, $\alpha_1 = 5.724$, and $\alpha_2 = -4.906$ for the inner tube, and reduced polynomial $N = 1$ (i.e., neo-Hookean) with $C_{10} = 0.121$ for the outer tube.

Next, we designed and assembled each component of the STV using a 3D modeling tool (Rhinoceros 7). For simplicity, the connectors were excluded from the design. The parts were modeled using a solid element type of C3D8R, and the fitted material coefficients were appropriately applied for each component. Thereafter, the WHY was tie-constrained to the inner tube and then inserted into the outer tube. Finally, to approximate the quasi-static loading conditions with negligible inertial effects (strain rate <0.4 s$^{-1}$)[60] while avoiding unnecessary simulation computational costs, the assembly was stretched at a strain rate of <0.025 s$^{-1}$. The elastic strain energy density $W$ was derived post-simulation by averaging the SENER field output of the inner tube elements located in the mid-cross-sectional plane.

Our simulation model provides easy investigation and insights into the effect of geometrical factors on STV behavior. The morphing behavior for various $n$, $p$, and $L_O$ values is shown in Supplementary Movie 2.

## Computational fluid dynamics

Since the analytical solution to the pressure drop in a deformed channel is not trivial, we coupled our finite element mechanical simulations with CFD simulations to predict the output chamber pressure curves and aid future design.

We exported the morphed surface data at multiple extension points from the FEA results (see Methods section 'Finite element simulations') and then reconstructed it with a retopology algorithm quad-remesher (Rhinoceros 7) to create manageable polygon meshes from the existing surfaces. We accounted for the load caused by the supply pressure, which expanded the outer tube (the inner tube had negligible effect), by 3D scaling up the deformed outer tube geometry proportionately (by a factor of 1.022 at 60 kPa, see Supplementary Fig. 11e). Then, we extracted the topology of the inlet and outlet channels and connected their ends with a plug using Boolean commands (Supplementary Fig. 2a). The geometry was then imported into ABAQUS and modeled using a fluid element of type FC3D4 and incompressible transient air with the following parameters: density $\hat{\rho} = 1.184$ kg/m$^3$ and dynamic viscosity $\bar{\mu} = 1.849 \times 10^{-5}$ kg/m·s. The flow inside the channels with a low Reynolds number is governed by the incompressible Stokes equation as follows:

$$-\nabla \bar{p} + \bar{\mu} \nabla^2 u = 0, \nabla \cdot u = 0, \tag{8}$$

where $\bar{p}$ is the pressure and $u$ is the velocity vector. We defined the inlet with $\bar{p} = P_s$, the outlet with $\bar{p} = 0$, and the surface of the channels with no-slip boundary conditions ($u = 0$). Finally, a steady-state solution of Eq. (8) was achieved using a backwards Euler method as follows:

$$\frac{du_{n+1}}{dt} = \frac{u_{n+1} - u_n}{\triangle t}. \tag{9}$$

The pressure profile throughout the channels at different extension points is shown in Supplementary Fig. 2b.

## Soft gripper

The soft gripper was fabricated by integrating a soft pneumatic actuator, a CO2 canister (95 g; UP), a mechanical pressure regulator (IR2000-02G-A; SMC), and an STV into a soft-foamed body. Details specific to the materials and the fabrication of the soft pneumatic actuator are provided in our previous work[61]. Briefly, Dragonskin 30 A (Smooth-On) was used as the soft gripper material, produced through conventional molding process using a 3D printer (Stratasys). Then, rigid structures (ABS-P430; Stratasys), manufactured via 3D printing, were inserted into punched holes of the DragonSkin 30 A endoskeleton and secured.

## Soft exosuit

The soft exosuit was fabricated first by stitching a pocket and cuffs to a commercially available jacket. Then, a CO$_2$ canister (95 g; UP) and a mechanical pressure regulator (IR2000-02G-A; SMC) were placed in the pocket. The inlet-outlet connector of the STV, a routing ring, and a soft elbow actuator were fixed to the cuffs. The soft elbow actuator parts were molded out of Dragon Skin 30 (Smooth-On) and assembled using a silicone adhesive (Silpoxy; Smooth-On). Finally, a tendon was anchored to the chamber connector and the routing ring.

For characterization, we attached a motor (H54-200-S500; Robotis) and a torque sensor (RFT64-SB01) to the elbow of the mannequin and measured the torque applied to the elbow every 5°. The weight of the elbow was 1.6 kg, similar to that of an average man with a height of 173.7 cm and a weight of 76.5 kg[62]. The reduction in gravitational torque acting on the elbow using the exosuit with STV compared to the undressed condition $\tau_r$ was calculated using the following equation:

$$\tau_r = \int_0^{60°} \left| \frac{\tau_g - \tau_{STV}}{\tau_g} \right| d\theta, \tag{10}$$

where $\tau_g$ and $\tau_{STV}$ are the torques applied to the elbow when the mannequin is undressed and dressed in the exosuit with STV control, respectively.

**Reporting summary**

Further information on research design is available in the Nature Portfolio Reporting Summary linked to this article.

## Data availability

All data generated or analyzed during this study are included in the published article and its Supplementary Information and are available from the corresponding author on request.

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

## Acknowledgements

This work was supported by National Research Foundation of Korea (NRF) grants funded by the Korean government, 2017K1A4A3015437 (J.K.), NRF-2020R1A2C2102842 (J.K.), NRF-2021R1A4A3033149 (J.K.), NRF-2019R1A2C2084677 (J.B.), RS-2023-00208052 (J.B.) and NRF-2019-Global Ph.D. Fellowship Program (J.K.C.), the Fundamental Research Program of the Korea Institute of Material Science, PNK7630 (J.K.), the research fund of UNIST, 1.220052.01 (J.K.), and Korea Evaluation Institute of Industrial Technology (KEIT) grant funded by the Korea Government, No. 20008912 (J.B.).

## Author contributions

J.K.C., J.-S.K., J.B., and J.K. conceived the project. J.K.C., J.-S.K., and H.S. conducted the experiments. J.K.C., J.-S.K. analyzed the data. J.K.C. performed the finite element and computational fluid dynamics simulations. J.K.C., J.-S.K. wrote the manuscript with input from all authors. J.B., J.K. supervised the study.

## Competing interests

The authors declare no competing interests.
