## [Peer Review File · Nature Communications]

REVIEWER COMMENTS

Reviewer #1 (Remarks to the Author):

The authors presented an intelligent and interesting way to produce a self-sensing tensile valve (STV) for soft robotics. Noticeably, STVs were obtained by an agile manufacturing process and were completely electronics-free. Furthermore, the authors developed an FEA model to program and predict pressure curves fitting well with the experimental results. These devices are expected to have a relevant application in several sub-fields of soft robotics, such as underwater and in vivo manipulation.

The article has good scientific sounding, and it is clear. Perhaps the clearness of the figures may be improved. Moreover, according to this reviewer, the methodology can still be improved to understand the proposed devices' reliability and reproducibility.

In the proposed paper, always as a boundary condition, the maximum strain is reported, but its numerical value is not registered. I suggest writing this value to improve comprehension.

In my opinion, the methodology applied for FEA and experimental tensile testing is not very clear. Starting from FEA, in page 13 it is reported that uniaxial testing was performed for the inner and the outer tube at 2mm/min. After finding the hyperelastic model, in page 14 it is reported that the components were assembled via a 3D modeling tool and stretched at a speed of 1mm/s, always using FEA.

- Why were the materials and the final assembly tested with such different strain rates (2mm/min vs 1mm/s)?
- Were real STV devices tested and compared with the FEA model for validation? Under which conditions?
- The authors in general report only 1 loading curve at a constant strain rate. Did they also characterize the device for different strain rates and cyclic loading and unloading conditions? Did they observe strain dependence or hysteresis?
- Regarding reliability, how many samples were tested for each characterization?

Other small issues are the following

- To improve the understanding of the work, I suggest adding in figure 2b the transversal view of the devices at the increasing number of WHY (n).
- In page 9 I would explicit that the handle of the gripper was pulled manually. Did the authors elaborate on a more automatic way of actuation?
- Page 13, line 324, the authors probably wanted to refer to Figure 7b?

Reviewer #2 (Remarks to the Author):

This work presents a self-sensing tensile valve (STV) which leverages helical pinching to enable tensile strain to be converted into an output pressure using only one single pressure source. In addition to self-sensing the valve can also allow proportional control of a soft pneumatic actuator from a single, constant supply pressure. The structure of valves incorporates wrapping helical yarns around an inner tube that when stretched they deform the inner chamber changing the cross-sectional area and hence airflow. The authors present FEA and CFD analysis and analytical models of these helical pinching systems.

I found the highlight of this paper to be integration of the helical yarns into the inner tube to provide the mechanical regulation of flow rate. This is a highly elegant problem to self-regulation of the pressure flow. This concept is systematically explored and illustrated through simulation and experiment. The ability to use the wrapped helical yarn to program the pressure response further demonstrates the capabilities of this approach.

Whilst the development of this self-sensing regular shows potential, I think this paper does not sufficiently exploit the concept of the geometric sensing regulation through helical yarn wrapping (e.g. as a wider theory) opposed to just valve control which would have more significance towards the larger community. Thus, although an interesting concept, I do not find the theoretical or application oriented contributions to be sufficient to recommend acceptance in nature communications at this time.

The authors position the paper as making contributions to electronics free soft robots. I think the authors could make improved comparison to electronics free soft robots and control; there are some notable examples of cyclic motions and also electronics free tactile sensing leveraging valves. How this work sit in this competitive area, in particular in comparison to microfluidic systems and microfluidic logic. To me it is not clear the advantages of this approach in comparison to the existing state of the art in this area.

In addition, I have some smaller comments. Helical tendon routing has been explored in the role of action (e.g. to generate twisting motions), adding in references and comparison to this approach to previous exploitation of helical wrapping could be a good addition.

The ability to tune the sensitive mechanically is interesting, however, I wonder as to the range that is achievable – it seems like the range in Fig 2C is not huge – some context at to the resulting effects of the different sensitivity (in a quantitative manner) would better demonstrate the role for of the mechanically programmable sensitivity. I find the grasping results in Fig 3 lack some rigour and quantative representation; what was the success rate, how did the sensitivity help/assist the task and how was it chosen for the task.

The electronics-free soft gripper highlights the ability to regulate the inlet and outlet valve, and the tuning seems to allow grasping of delicate and high force objects. I find it hard to see the significant advance of this application in comparison to other control methods for soft actuator control, as this could essentially also be achieved by a human triggering a valve, or pulling on tendons.

I am unsure as to the motivation behind the CFD simulations. The pneumatic systems is relatively simple (i.e. in terms of the volume/flow rate), are there any more complex flow interactions (vortices) or other behaviours the authors expected to motivate CFD analysis?

Reviewer #3 (Remarks to the Author):

Please refer to attached review document.

Review of “A Soft, Self-Sensing Tensile Valve for Perceptive Soft Robots” (NCOMMS-22-37267) by Jun Kyu Choe, Junsoo Kim, Hyeonseo Song, Joonbum Bae, and Jiyun Kim

In this manuscript, the authors describe a soft pneumatic valve that produces an analog pressure output in response to tensile strain. This strain-sensing valve, termed an STV by the authors, is constructed from a pair of concentric elastomeric tubes that act as flow conduits. A helically wrapped inextensible yarn pinches the inner tube and generates a space between the inner and outer tubes when stretched, and the corresponding changes in flow resistance enable the STV to behave as a fluidic voltage divider, producing an output pressure that changes in response to applied strain. The authors demonstrate the use of their device in controlling a soft gripper and an elbow assist device, both of which respond to mechanical strain applied to the STV.

The manuscript is interesting and represents a timely contribution to the ongoing effort in soft robotics toward developing soft sensors and transducers that directly output fluidic pressure signals. However, additional details on the performance and characteristics of the valve, as outlined in the subsequent comments, must be included to provide a more complete picture of its capabilities and limitations. This expansion of detail will considerably aid in customization and deployment of the authors’ approach in future soft robots and wearable devices. Furthermore, there has been considerable research interest in developing soft sensors—including pneumatic strain sensors—with several papers published in recent years; the advantages and limitations of the authors’ presented device needs to be more clearly articulated in comparison with these earlier approaches. Specific questions and comments for the authors are listed below.

1. The reference list has omitted several related relevant works. The authors are encouraged to cite recent literature on fluidic sensors and supporting soft and fluidic control infrastructure relevant to their work. A few examples, mostly from 2022, are listed below (this list is not exhaustive; the authors are encouraged to search for additional relevant work):

- [1] Hevia, et al., “High-Gain Microfluidic Amplifiers: The Bridge between Microfluidic Controllers and Fluidic Soft Actuators,” *Advanced Intelligent Systems* (2022).
- [2] Davletshin, et al., “A Bidirectional Soft Diode for Artificial Systems,” *Advanced Functional Materials*, 2200658 (2022).
- [3] Rajappan, et al., “Logic-Enabled Textiles,” *Proceedings of the National Academy of Sciences*, 119, 35 (2022).
- [4] van Laake, et al., “A Fluidic Relaxation Oscillator for Reprogrammable Sequential Actuation in Soft Robots,” *Matter*, 5, 9 (2022).
- [5] Decker et al., “Programmable Soft Valves for Digital and Analog Control,” *Proceedings of the National Academy of Sciences*, 119, 40 (2022).
- [6] Koivikko et al., “Integrated Stretchable Pneumatic Strain Gauges for Electronics-Free Soft Robots,” *Communications Engineering* 1, 14 (2022).
- [7] Jin, et al., “Mechanical Valves for On-Board Flow Control of Inflatable Robots,” *Advanced Science*, 8, 2101941 (2021).

- [8] Song, et al., “CMOS-Inspired Complementary Fluidic Circuits for Soft Robots,” *Advanced Science*, 8, 2100924 (2021).
- [9] Sourì et al., “Wearable and Stretchable Strain Sensors: Materials, Sensing Mechanisms, and Applications,” *Advanced Intelligent Systems*, 2, 2000039 (2020).
- [10] Yeo et al., “Flexible and Stretchable Strain Sensing Actuator for Wearable Soft Robotic Applications,” *Advanced Materials Technologies*, 1, 1600018 (2016).
- [11] Mengüç, et al., “Wearable soft sensing suit for human gait measurement,” *The International Journal of Robotics Research*, 33, 1748 (2014).

2. Regarding the omitted references mentioned in the previous comment, reference [5] demonstrates an analog pressure regulator that produces an output pressure proportional to force input from a human user. If the system described in this work were attached to a rubber band to convert strain to applied force, it seems it would generate essentially the same output as the authors’ submitted work. The distinction from prior work or advantages of the present work should be emphasized.

3. Reference [6] above describes a pneumatic strain sensor that also uses a fluidic resistance-based voltage divider, similar to the approach taken in this work. Again, the distinction from prior work or advantages of the present work should be emphasized.

4. In the abstract, the authors state that the STV “*integrates the functional capabilities of sensors, controllers, and pneumatic valves*” (lines 30–31) and realizes “*physical sharing of both sensing and control structures*” (line 33). It is not clear if the current device can be regarded as incorporating any “control” capabilities, as it functions only as a strain sensor, i.e., a transducer that converts a mechanical input signal to a pressure output signal. For example, a strain bridge (which acts as an electronic voltage divider that also responds to strain) may be used in conjunction with a voltage source to provide an analog voltage output that could, in principle, drive an electrical actuator; however, it is still regarded only as a sensor, and not as an analog controller. Therefore, the aforementioned claim should either be moderated or properly justified.

5. At intermediate output pressures (when the inlet and outlet channels are partially open), the STV incurs a continuous loss of pressurized gas from the supply source to the atmosphere, which could pose a serious limitation when using finite and exhaustible onboard gas sources (such as a gas canister). Could the authors quantify the gas loss through their device as a function of the level of strain, and also comment on the longevity of their portable demos?

6. Neither the finite element analysis nor the analytical model presented in the manuscript seems to account for loads arising from differential gas pressure across the tube walls, and no justification is provided for their exclusion. Do forces arising from internal gas pressure inflate the tubes or otherwise influence their deformation profile? Does varying the supply pressure change the input-output response of the STV? It might be that pressure forces can be neglected due to the small diameter of the tubes used; if this is indeed the case, can the authors provide a

suitable quantitative scaling to show that the pressure forces are much smaller than the contact forces exerted by the yarns?

7. A key characterization that is missing is the input force required to stretch the STV from zero to full extension, which can offset some of the assistive effort provided by the actuator in wearable assistive applications. There is currently only a passing mention of the magnitude of this force on line 227, where the authors state that it is of the order of 10 N, which seems non-negligible. A force vs elongation curve for the STV should be straightforward to obtain using the authors' stated experimental capabilities. Does this force curve change with the supply pressure?

8. Although designed to be operated primarily under tension, it is conceivable that the STV might experience occasional compressive loads in wearable applications. How does the device fare under compression, buckling, bending, or kinking during routine use?

9. Scalability, cost, and robustness are all key considerations for soft valves intended for wearable applications. Could the authors provide more information on the durability of their STVs, e.g., the maximum supply pressure or the number of stretch cycles they can withstand before failure? Can the valves be repaired if damaged in use? How costly are they to manufacture, and can the fabrication process be scaled up for mass-production?

Other minor comments:

10. In Fig. 3g, the authors show their soft gripper handling a lit sparkler to demonstrate the advantages of its electronics-free construction and suitability for environments involving sparks (line 214). This is not a relevant demonstration, as electronic circuits are affected by electromagnetic interference caused by *electrical sparks*, not those produced by chemical combustion. A demonstration of the device operating near a spark gap, welding arc, or an automobile spark plug would be more appropriate in this context.

11. Were the elongation vs output pressure curves in Fig. 2 measured during both elongation and retraction of the STV? Given the buckling-dominated behavior for $n = 2$ and 4 yarns, it is possible that the valve response curve might show discernible hysteresis.

12. On line 22, the formula for the flow rate is stated as $Q = 15\pi vd^2$, whereas it should correctly read $Q = (1/4)\pi vd^2$. This could be a simple typographical error, but the authors should confirm that the correct numerical coefficient was used in the actual calculations, as it could significantly affect the determination of the Reynolds number and, by extension, the flow regime.

13. The depiction of intervals with negative pressure (outward arrows) in Extended Data Fig. 4 is inconsistent with the functional form shown in Eq. S1, which is always non-negative.

Response to reviewers' comments for the manuscript – “A Soft, Self-Sensing Tensile Valve for Perceptive Soft Robots”

Response to reviewer #1's comments

General Comment: The authors presented an intelligent and interesting way to produce a self-sensing tensile valve (STV) for soft robotics. Noticeably, STVs were obtained by an agile manufacturing process and were completely electronics-free. Furthermore, the authors developed an FEA model to program and predict pressure curves fitting well with the experimental results. These devices are expected to have a relevant application in several sub-fields of soft robotics, such as underwater and in vivo manipulation.

Response for General Comment: We appreciate the reviewer's valuable comments and favorable evaluation of our manuscript. All comments are very valuable and helpful for the improvement of our paper. We have carefully addressed each comment with point-by-point replies as follows.

Comment 1-1: The article has good scientific sounding, and it is clear. Perhaps the clearness of the figures may be improved.

Response 1-1: The authors thank the reviewer's favorable evaluation and comment. We added detailed descriptions to clarify any ambiguous part of the manuscript, and modified figures to improve the reader's understanding based on the reviewer's comments (please refer to **Response 1-3, Response 1-8**).

Comment 1-2: Moreover, according to this reviewer, the methodology can still be improved to understand the proposed devices' reliability and reproducibility.

Response 1-2: The authors thank the reviewer's comment. Based on the reviewer's comments, we did experiments and added detailed descriptions for the better understanding of the STV's reliability and reproducibility (please refer to **Response 1-6, Response 1-7**).

Comment 1-3: In the proposed paper, always as a boundary condition, the maximum strain is reported, but its numerical value is not registered. I suggest writing this value to improve comprehension.

Response 1-3: We appreciate the reviewer's excellent suggestion. We have added the numerical value in Supplementary Fig. 4c, 4e, and 4g in page 10 of “Supplementary information” file as follows:

Supplementary Fig. 4. Supplementary analysis results of the STV. ... c Normalized chamber pressure P_{ch}/P_s plotted against the extension length ΔL for different numbers of WHYs n ; $n = 1$ ($\epsilon_{max} = 0.46$, $S = 3.34$), $n = 2$ ($\epsilon_{max} = 0.32$, $S = 17.26$), $n = 3$ ($\epsilon_{max} = 0.27$, $S = 7.71$), and $n = 4$ ($\epsilon_{max} = 0.21$, $S = 40.66$). ... **e** Normalized chamber pressure P_{ch}/P_s plotted against the extension length ΔL for different cyclic pitch p values; $p = 10$ mm ($\epsilon_{max} = 0.32$), $p = 15$ mm ($\epsilon_{max} = 0.15$), and $p = 20$ mm ($\epsilon_{max} = 0.09$). ... **g** The maximum strain ϵ_{max} increases and then saturates as L_0 increases. The dotted line represents the analytical result (see Supplementary Note 2). The numerical values of ϵ_{max} for different L_0 are as follows: $L_0 = 40$ mm ($\epsilon_{max} = 0.27$), $L_0 = 60$ mm ($\epsilon_{max} = 0.29$), $L_0 = 80$ mm ($\epsilon_{max} = 0.32$), $L_0 = 100$ mm ($\epsilon_{max} = 0.35$), and $L_0 = 120$ mm ($\epsilon_{max} = 0.37$). ...

Comment 1-4: In my opinion, the methodology applied for FEA and experimental tensile testing is not very clear. Starting from FEA, in page 13 it is reported that uniaxial testing was performed for the inner and the outer tube at 2mm/min. After finding the hyperelastic model, in page 14 it is reported that the components were assembled via a 3D modeling tool and stretched at a speed of 1mm/s, always using FEA.

Why were the materials and the final assembly tested with such different strain rates (2mm/min vs 1mm/s)?

Response 1-4: The tensile test was conducted to find the mechanical properties of the materials, and the FEA was conducted to find the inlet and outlet channel geometries under extension.

Since STVs intend to achieve continuous stable states to target specific stabilized chamber pressures, we first intended to find the channel geometries at static extension states. To avoid the inertial effects at high strain rates which could change the channel geometries, we arranged the tensile testing and FEA to achieve quasi-static loading conditions ($< 0.4 \text{ s}^{-1}$ [R1]). Specifically, the uniaxial testing we performed at the extension speed of 2 mm/min (0.033 mm/s) can be normalized given the initial length of the dog-bone shape samples (54 mm). This result in the strain rate of 0.0006 s^{-1} , which is under the quasi-static loading condition. In FEA, we increased the extension speed (1 mm/s) to reduce the computational cost. We conducted the FEA with the extension speed of 1 mm/s (with initial length of $40 \text{ mm} < L_0 < 120 \text{ mm}$, yielding $0.008 \text{ s}^{-1} < \text{strain rate} < 0.025 \text{ s}^{-1}$) which is also under the quasi-static loading condition. For clarification, we revised the inconsistent usage of units (from (mm/min) and (mm/s) to (s^{-1})) and revised the “Finite Element Simulations” section in page 13 and 14 as follows:

“First, to obtain the elastic property coefficients, we performed uniaxial tension tests (AGX-100NX; SHIMADZU) of the WHY, inner tube, and outer tube at a strain rate of 0.0006 s^{-1} .”

...

“Finally, to approximate the quasi-static loading conditions with negligible inertial effects (strain rate $< 0.4 \text{ s}^{-1}$)⁶⁰ while avoiding unnecessary simulation computational costs, the assembly was stretched at a strain rate of $< 0.025 \text{ s}^{-1}$.”

[R1] M. Cheng, W. Chen, Experimental investigation of the stress–stretch behavior of EPDM rubber with loading rate effects. *International Journal of Solids and Structures* **40**, 4749-4768 (2003).

Comment 1-5: Were real STV devices tested and compared with the FEA model for validation? Under which conditions?

Response 1-5: We appreciate the reviewer’s valuable comment. For the validation, we compared the cross-sectional images obtained from the micro-CT experiment and the FEA at the mid-position of the STV (Fig. 2b). Then we derived analytical expressions (Supplementary Fig. 7, Supplementary Fig. 8, and Supplementary Note 1, 2) which showed good agreement. Then we further implemented the CFD model (Supplementary Fig. 2) to compare the resulting simulation and experimental chamber pressures at each extension level of the STV, which also showed good agreement (Fig. 1d).

For the characterization, we tested the STV under steady-state conditions as the real STV devices intend to achieve continuous steady-state pressure states depending on the strain. Specifically, we extended the STV 0.1 mm at a time and then held the position for 0.4s after steady-state was reached and collected data using the characterization set up in Supplementary Fig. 11. Similarly, the FEA tests were designed with low strain rates that avoid the inertial effects (please refer to **Response 1-4** for more detail). Finally, the CFD was conducted to achieve the steady-state solution of the incompressible Stokes equation Eq. (8).

For clarification, we revised the “Abstract” section in page 2 as follows:

“Here, we report a soft self-sensing tensile valve that integrates the functional capabilities of sensors, embedded controllers, and pneumatic valves to directly transform applied tensile strain into distinctive steady-state output pressure states using only a single constant pressure source.”

Also, we revised the “Analog Control of Pneumatic Actuation” section in page 4 and 5 as follows:

“The STV transforms the applied strain into distinctive steady-state output chamber pressure states.”

...

“The experimental data obtained at steady-state conditions (see Methods section ‘soft tensile valve characterization’) and CFD simulation show good agreement, demonstrating that our model is accurate.”

Comment 1-6: • The authors in general report only 1 loading curve at a constant strain rate. Did they also characterize the device for different strain rates and cyclic loading and unloading conditions? Did they observe strain dependence or hysteresis?

Response 1-6: The authors thank the reviewer’s incisive comments. We have carefully addressed each question with point-by-point replies as follows:

Response 1-6.1: Strain rate.

Although the STVs are intended to produce output pressure states under steady-state conditions (please refer to **Response 1-5**), the output chamber pressure is a strain rate-dependent function due to the intrinsic limitation in the speed of air. Particularly, extending the STV at faster speeds could result in reduced output chamber pressures if the air from the inlet does not have enough time to move into the actuators to achieve the target chamber pressure state. Likewise, contracting the STV at higher speeds could result in increased output chamber pressures if the air from the actuator does not have enough time to exhaust. We redid the experiment at steady-state condition, strain rate of 0.0109 s^{-1} , and 0.0150 s^{-1} with the initial actuator volume of $39,919 \text{ mm}^3$. Note that these strain rates produce quasi-static channel geometries (without inertial effects), and thus the delay is mainly due to the limitation in the flow rate Q . The time required for the pressure to saturate for strain rate of 0.0109 s^{-1} and 0.0150 s^{-1} at $\varepsilon = \varepsilon_{\max}$ was 0.3 s and 1 s respectively. Note that the saturation time could be reduced by using an actuator with a lower initial actuator volume. We have added the data in Supplementary Fig. 5a in page 11 of “Supplementary information” file as follows:

Supplementary Fig. 5. Considerable factors of the STV during practical use. a Strain rate $\dot{\varepsilon}$ dependence of the STV. (i) schematic illustration of the STV test set-up (ii) P_{ch}/P_s plotted against the normalized strain $\varepsilon/\varepsilon_{\max}$ for different strain rates $\dot{\varepsilon}$. (iii) When $\varepsilon/\varepsilon_{\max} = 1$ is reached, the extension of the STV was maintained to allow the pressure inside the actuator to saturate to the target pressure. The saturation time is 0.3 s for $\dot{\varepsilon} = 0.0109$ and 1 s for $\dot{\varepsilon} = 0.0150$ ($n = 2$, $p = 10 \text{ mm}$, $L_0 = 100 \text{ mm}$). ...

Also, we added a discussion in Supplementary Note 3.1 in page 5 of “Supplementary information” file as follows:

3.1 Strain rate dependence

“The output chamber pressure is a strain rate-dependent function due to the intrinsic limitation in the speed of air. As shown in Supplementary Fig. 5a, extending the STV at faster speeds could result in reduced output chamber pressures if the air from the inlet does not have enough time to move into the actuators to achieve the target chamber pressure state. Likewise, contracting the STV at higher speeds could result in increased output chamber pressures if the air from the actuator does not have enough time to exhaust. In our system, the time required for the pressure to saturate for strain rate of 0.0109 s^{-1} and 0.0150 s^{-1} at $\varepsilon = \varepsilon_{\max}$ was 0.3 s and 1 s respectively. Note that the saturation time could be reduced by using an actuator with a lower initial actuator volume.”

Response 1-6.2: Cyclic loading and unloading.

Under cyclic loading and unloading, we found negligible hysteresis when $n = 1, 3$, and hysteresis when $n = 2, 4$. We found larger hysteresis when $n = 4$ than when $n = 2$. We have added the above information in Supplementary Fig. 4a in page 10 of “Supplementary information” file as follows:

Supplementary Fig. 4. Supplementary analysis results of the STV. a Cyclic loading and unloading results of the STV shown in Fig. 2c. ...

Also, we revised the “Pressure curve profile programming” section in page 6 as follows:

“Fig. 2c and Supplementary Fig. 4a show these curves in normalized form (P_{ch}/P_s) as a function of the normalized axial tensile strain ($\varepsilon/\varepsilon_{\max}$) applied to the STV.”

Comment 1-7: • Regarding reliability, how many samples were tested for each characterization?

Response 1-7: Regarding fabrication reliability, two samples were tested for each characterization. While an unautomated fabrication process inevitably creates variability in

function, we found the simple and agile manufacturing process of the STV created negligible deviation in output data before moving on to create more identical samples. However, we redid the experiment with 5 samples and found good reliability in reproducibility. We added this information in Supplementary Fig. 5c in page 11 of “Supplementary information” file as follows:

Supplementary Fig. 5. Considerable factors of the STV during practical use. ... P_{ch}/P_s plotted against the normalized strain $\varepsilon/\varepsilon_{max}$ with five different samples ($n = 2, p = 10 \text{ mm } L_0 = 120 \text{ mm}$). ...

Also, we added a discussion in Supplementary Note 3.3 in page 6 of “Supplementary information” file as follows:

3.3 Reproducibility

“While an unautomated fabrication process inevitably creates variability in function, we found the simple and agile manufacturing process of the STV created negligible deviation in output data when tested with 5 different samples (Supplementary Fig. 5c).”

Comment 1-8: Other small issues are the following. To improve the understanding of the work, I suggest adding in figure 2b the transversal view of the devices at the increasing number of WHY (n).

Response 1-8: We appreciate the reviewer’s excellent suggestion. To improve the understanding of our work, we added a new figure of the transversal view of the devices in Supplementary Fig. 3 in page 9 of “Supplementary information” file as follows:

Supplementary Fig. 3. FEA results with transversal cross-sectional views at the increasing number of n . a $n = 1$, b $n = 2$, c $n = 3$, and d $n = 4$.

We also revised the “Pressure Curve Profile Programming” section in page 6 as follows:

Fig. 2b shows FEA cross-sectional deformation results from $n = 1$ to $n = 4$ with the contours of the normalized elastic strain energy density W/W_{max} (see Methods section ‘Finite element simulations’) and characterization results by microcomputed tomography (micro-CT) (see Methods section ‘Microcomputed tomography’). For the transversal view of the FEA results, see Supplementary Fig. 3. Upon tensile strain, the WHYs are gradually straightened to be aligned on the central axis of the undeformed inner tube until mechanically confined by the inner tube.

Comment 1-9: In page 9 I would explicit that the handle of the gripper was pulled manually. Did the authors elaborate on a more automatic way of actuation?

Response 1-9: The authors thank the reviewer’s comments. The actuation of the gripper could potentially be achieved in a more automatic way. For example, implementing pantograph-based mechanical linkages to the gripper could allow transfer of objects’ reactive force into the

extension of the STV, and in tandem with multi-stable mechanisms, the gripper could create appropriate holding force depending on the object's reactive force to grasp various objects.

However, as we focused our demonstrations on human-robot interactions (HRI), we have not elaborated on a more automatic way of actuation in this paper. For clarification, we revised the “Untethered and Electronics-free Soft Gripper” section in page 9 as follows:

“Then, each connector of the STV was fixed to the body and the handle so that the STV could be tensioned by manually pulling the handle (Fig. 3c).”

Comment 1-10: Page 13, line 324, the authors probably wanted to refer to Figure 7b?

Response 1-10: We apologize for the confusion. Our original intention was to refer the stress-strain data to both Figure 7a and Figure 7b. While both Figure 7a and Figure 7b are stress-strain data, we plotted these figures separately for comprehensibility as the WHY and the elastomeric tubes have different orders of magnitude in stress. For clarification, we revised the “Finite Element Simulations” section in page 13 and page 14 in the direction of mentioning the figures separately as follows:

“The nominal stress-strain data of the WHY is shown in Supplementary Fig. 10a, and the nominal stress-strain data of the inner and outer tubes are shown in Supplementary Fig. 10b.”

Note that the name of the Supplementary Fig. 7 has been substituted to Supplementary Fig. 10.

Response to reviewer #2's comments

General Comment: This work presents a self-sensing tensile valve (STV) which leverages helical pinching to enable tensile strain to be converted into an output pressure using only one single pressure source. In addition to self-sensing the valve can also allow proportional control of a soft pneumatic actuator from a single, constant supply pressure. The structure of valves incorporates wrapping helical yarns around an inner tube that when stretched they deform the inner chamber changing the cross-sectional area and hence airflow. The authors present FEA and CFD analysis and analytical models of these helical pinching systems.

I found the highlight of this paper to be integration of the helical yarns into the inner tube to provide the mechanical regulation of flow rate. This is a highly elegant problem to self-regulation of the pressure flow. This concept is systematically explored and illustrated through simulation and experiment. The ability to use the wrapped helical yarn to program the pressure response further demonstrates the capabilities of this approach.

Response for General Comment: We thank the reviewer's valuable comments and helpful advice for our manuscript. As the reviewer pointed out, the integration of the helical yarns into the inner tube, followed by the integration of the outer tube into a core-shell structure provides the mechanical regulation of flow rate both in and out of a soft actuator in a programmable and compact manner.

Based on the reviewer's comments, we tried to improve our manuscript as best we could especially by clarifying the advantages or capabilities of the STV over other similar technologies. We addressed each comment with point-by-point replies as follows.

Comment 2-1: Whilst the development of this self-sensing regular shows potential, I think this paper does not sufficiently exploit the concept of the geometric sensing regulation through helical yarn wrapping (e.g. as a wider theory) opposed to just valve control which would have more significance towards the larger community. Thus, although an interesting concept, I do not find the theoretical or application oriented contributions to be sufficient to recommend acceptance in nature communications at this time.

Response 2-1: The authors thank the reviewer's valuable comments.

Soft robots have advantages over rigid robots in human-robot interactions (HRI), due to their flexible and compliant nature that allows for safe interactions with humans. Since humans have flexible and soft curvature and their motions and forces are all continuous, soft devices with compact form, stretchability, and continuous regulation capability have been spotlighted in HRI when sensing and utilizing humans' strain or force, as well as integrating with human bodies. However, in widely used pneumatic/hydraulic-based systems, many existing valves relied on electronic sensors and controllers, sacrificing compactness and softness (Fig. X1, [R2]). In the case where the soft valves make self-regulation in response to external mechanical stimuli,

these valves are not sufficiently compliant [R3], have a bulky form factor [R4] or lack sensing capabilities to fully utilize proportional strain or force created by humans [R5]. Therefore, to effectively use the advantages of soft robots in HRI, a new control device should self-regulate its geometry to combine pressure control with proportional sensing of external stimuli, within soft and compact form factors.

Fig. X1. Soft exosuit using electronic valve and controller for regulation [R2].

The STV mechanically embeds the functions of soft stretchable sensors and electric valves into its fully soft, linear, and compact form factor, transducing self-sensed proportional strain into output pressures to realize electronics-free continuous control of soft robots. To the best of our knowledge, the concept of utilizing the geometry of helical yarns has been limited to strain sensors [R6] and actuators [R7-9] in the field of soft robotics (also, please refer to **Response 2-3**), despite the fact that it can create stretchable linear form factor devices that offer some functional and integrational advantages in HRI. We first expanded the use of this geometrical concept by creating a mechanism for the control of soft robots. Our helical pinching mechanism enables compact, linear, and fully soft designs of the STV with tunable proportional control capabilities from a single, constant input pressure, which is difficult to achieve using conventional valve control approaches. Later, we present a comparison of the advantages of the STV vs existing valves (Supplementary Table 1). Also, we derived analytical models for our system and numerical models which could aid future theoretical designs of the STV.

The most beneficial application for soft robots when utilizing responses to continuous strain or stimuli is HRI. This is because humans have flexible and soft curvature, and their motions and forces are all continuous. However precise and intuitive regulation or synchronization of soft robots to human movement is possible only when the proportional strain is read and utilized accordingly. Therefore, to demonstrate the novelty of the STV, we showed two HRI applications that distinguish the STV from other soft valves (a gripper that uses human motion for precise and intuitive regulations, and an exosuit that reads human motion and autonomously synchronizes its pressure states). Particularly, soft exosuits have been in the spotlight for rehabilitation and injury prevention, yet there were concerns regarding complexity, high cost, safety, and discomfort due to rigid components that rely on rigid electronics. We demonstrated that a soft exosuit with the STV may easily alleviate these problems.

Furthermore, we show that digital regulation and oscillation capabilities, which are mainly demonstrated by existing soft valves, are all achievable (please refer to **Response 2-2**). Therefore, as a follow-up study, we plan to exploit this capability to read digitized strain and realize oscillations to manufacture an independent soft robot that show self-regulation capability for various stimuli.

[R2] Nassour, J., Zhao, G. & Grimmer, M. Soft pneumatic elbow exoskeleton reduces the muscle activity, metabolic cost and fatigue during holding and carrying of loads. *Sci. Rep.* **11**, 12556 (2021)

[R3] C. J. Decker *et al.*, Programmable soft valves for digital and analog control. *Proc. Natl. Acad. Sci. USA* **119**, (2022).

[R4] P. Rothemund *et al.*, A soft, bistable valve for autonomous control of soft actuators. *Science Robotics* **3**, eaar7986 (2018).

[R5] D. J. Preston *et al.*, Digital logic for soft devices. *Proc. Natl. Acad. Sci. USA* **116**, 7750-7759 (2019).

[R6] Lee, J. et al. Stretchable and suturable fibre sensors for wireless monitoring of connective tissue strain. *Nature Electronics* **4**, 291-301 (2021).

[R7] C. S. Haines *et al.*, New twist on artificial muscles. *Proc. Natl. Acad. Sci. USA* **113**, 11709-11716 (2016).

[R8] J. E. Slightam, M. L. Nagaruka, Theoretical Control-Centric Modeling for Precision Model-Based Sliding Mode Control of a Hydraulic Artificial Muscle Actuator. *Journal of Dynamic Systems, Measurement, and Control* **143**, (2021).

[R9] I. S. Yahara, S. Wakimoto, T. Kanda, K. Matsushita, McKibben artificial muscle realizing variable contraction characteristics using helical shape-memory polymer fibers. *Sensors and Actuators A: Physical* **295**, 637-642 (2019).

Comment 2-2: The authors position the paper as making contributions to electronics free soft robots. I think the authors could make improved comparison to electronics free soft robots and control; there are some notable examples of cyclic motions and also electronics free tactile sensing leveraging valves. How this work sit in this competitive area, in particular in comparison to microfluidic systems and microfluidic logic. To me it is not clear the advantages of this approach in comparison to the existing state of the art in this area.

Response 2-2: the authors thank the reviewer's valuable comments. As the reviewer mentioned, there are some notable examples of electronics-free soft robots and control.

The state-of-the-art cyclic motions [R10] and tactile sensing leveraging valves [R11,12] are realized based on digital on/off valves. While these digital valves are advantageous in facilitating logic gates or cyclic motions, they are difficult to implement for "proportional" applications (e.g., force control, and closed-loop pressure control) where continuous and precise control is required. For example, to develop a soft two-bit digital-to-analog converter with four different output pressure states ranging from 0 to maximum pressure, a complex

connection of 2 control, 4 constant pressure sources, and 12 valves were required [R11]. Furthermore, achieving higher-resolution analog control (with more output pressure states) via on/off switching valves is expected to be exponentially complex and bulky, and a truly analog valve cannot be achieved using digital valves. In addition, this approach requiring multiple pressure inputs to achieve analog control impedes the development of electronics-free systems.

Similarly, microfluidic systems are typically based on quake valves which require multiple pressure inputs as they are operated based on pinching of channels using control pressure sources. Furthermore, in general their physical dimensions result in slow flow rates that are not practically applicable in soft robotics where macroscopic motions and forces are required. On the other hand, the STV enables a truly analog control for perceptive robots. Only using soft materials, the STV is capable of self-sensing and proportional control from a single, constant input pressure which reinforces the way towards electronics-free soft robots.

In addition, while we intended to focus on analog control capability and tunability of the STV in this paper, achieving digital logic or cyclic motions with the STV is theoretically possible as we can increase the sensitivity of the pressure curve (e.g., $n = 4$) to achieve pressure curve similar to an on/off valve. Embedding the STV in a linear actuator and exploiting inverse mode that flips the pressure curve generated by the STV could possibly enable digital logic gates as shown in Fig. X2 and cyclic motions as shown in Fig. X3.

Fig. X2. Digital logic using STV. (a) NOT (original mode STV) and (b) AND logic gate (inverted mode STV).

Fig. X3. Cyclic motion using STV. (a) Initial condition inflating the right actuator and extends the right STV. **(b)** When the threshold of $P_{ch,R} = 1$ (ON) reaches, the left actuator is inflated extending the left STV. **(c)** When the threshold of $P_{ch,L} = 0$ (OFF) reaches, the right actuator is deflated. **(d)** When the threshold of $P_{ch,R} = 0$ reaches, the left actuator is deflated. **(e)** When the threshold of $P_{ch,L} = 1$ reaches, the right actuator is inflated again.

While it is difficult to compare soft valves in general since they are not standardized, we have added information about the STV in comparison to relevant works in Supplementary Table 1 based on our knowledge in page 19 of “Supplementary information” file as follows:

Capability	Our valves	Micro-Fluidics ^{23, 40}	Bistable silicone valves ^{4, 49, 50, 53}	Digital, analog combinative valves ⁵²	Bidirectional check valves ⁴¹	Sheet-based valves ⁴⁷	Hysteretic valves ⁴⁶	Stretchable strain gauge ³⁶
Mechanism	Helical pinching	Linear pinching	Kinking	Kinking	Leaflet Flapping	Kinking	Slits in soft membrane	Channel elongation
Device shape	Tubular	Rectangular/Sheet	Cylindrical	Tubular	Tubular	Sheet	Tubular	Rectangular
Characteristic scale	Diameter = 5 mm	Thickness = 1 ~ 2.8 mm	Diameter = 27 mm, height = 34 mm	Diameter = 15 mm*	Channel length = 4 ~ 8 mm	Thickness = 2 mm**	Diameter = 5 mm	92 x 15 x 4 (mm)***
Practical flow rates for actuation	✓	—	✓	✓	✓	✓	✓	—
proportional control capabilities	✓	—	—	✓	—	—	—	—
Programmable proportional control	✓	—	—	—	—	—	—	—
Operation with single, constant pressure source	✓	—	✓	✓	—	—	✓	—
Proportional self-sensing capabilities	✓	—	—	✓	—	—	—	✓
Inflow and outflow resistance control	✓	—	—	—	—	—	—	—
Entirely soft	✓	✓	✓	—	✓	✓	✓	✓

* Estimated from Fig. 1c in⁵²
**Estimated from Fig. S13 in⁴⁷
*** Estimated from Fig. S13 in³⁶

Supplementary Table 1. Comparison of our valves to recently developed soft control devices.

(We note the references in this table can be found in page 17-19 of the “Reference” section of the manuscript.)

[R10] Preston, D. J. *et al.* A soft ring oscillator. *Science Robotics* **4**, eaaw5496 (2019).

[R11] Preston, D. J. *et al.* Digital logic for soft devices. *Proc. Natl. Acad. Sci. USA* **116**, 7750-7759, doi:10.1073/pnas.1820672116 (2019).

[R12] A. Rajappan *et al.*, Logic-enabled textiles. *Proc. Natl. Acad. Sci. USA* **119**, e2202118119 (2022).

Comment 2-3: In addition, I have some smaller comments. Helical tendon routing has been explored in the role of action (e.g. to generate twisting motions), adding in references and comparison to this approach to previous exploitation of helical wrapping could be a good addition.

Response 2-3: The authors thank the reviewer’s valuable suggestions. As the reviewer mentioned, helical tendon routing has been previously explored in other devices, but we used

it for the first time in soft valves to create a compact valve with proportional self-sensing and control capabilities. We have added information about the STV in comparison to other devices that use helical routing structures in Supplementary Table 2 in page 20 of “Supplementary information” file as follows:

	This work	Ref. ⁶⁴	Ref. ⁶⁵	Ref. ⁶⁶	Ref. ⁶⁷	Ref. ⁶⁸	Ref. ⁶⁹	Ref. ⁷⁰
Materials	Polyethylene yarn/ rubber tube	Spandex fiber/ silicone elastomer	Nylon/ copper wire	Kevlar fiber/ silicone elastomer	Tendon/ dexterous manipulator	Polyamide fiber/ polyurethane elastomer	Cellulose acetate, PLGA, polyurethane Nanofiber	Gold nanowire- impregnated fiber/ fiber conductor
Type	Pneumatic valve	Sensor	Artificial muscle	Pneumatic actuator	Continuum manipulator	Auxetics	Micro tissue	Supercapacitor
Mechanism	Inflow & outflow pneumatic resistance tuning	Capacitive	Radial-axial thermal expansion anisotropy	Strain limitation	Strain limitation	Interactive normal forces	Coil opening	Coil opening
Function	Proportional self- sensing and control of pneumatic actuators	Strain sensing	High work capacity contraction	Twist motion	S-shape motion	Negative Poisson's ratio	High stretchability	High stretchability

Supplementary Table 2. Comparison of our work to other devices using helical routing structures.

(We note the references in this table can be found in “Supplementary Reference” section in page 23 of the “Supplementary information” file.)

Comment 2-4: The ability to tune the sensitive mechanically is interesting, however, I wonder as to the range that is achievable – it seems like the range in Fig 2C is not huge – some context at to the resulting effects of the different sensitivity (in a quantitative manner) would better demonstrate the role for of the mechanically programmable sensitivity.

Response 2-4: We appreciate the reviewer’s valuable comments. We calculated the sensitivity $S = \delta(P_{ch}/P_s)/\delta\varepsilon$, regarding that ε_{max} is different depending on the number of WHY(n) (Supplementary Fig. 4c), at the intermediate of the maximum P_{ch}/P_s ($P_{ch}/P_s = 0.5$ for $n = 2-4$, and $P_{ch}/P_s = 0.37$ for $n = 1$). We obtained 12-fold tunability in sensitivity ($S = 3.34$ for $n = 1$, $S = 17.26$ for $n = 2$, $S = 7.71$ for $n = 3$, and $S = 40.66$ for $n = 4$).

While some existing soft strain sensors show high tunability in sensitivity (e.g., 171-fold gauge factor tunability from 23 to 3933 [R13]), most strain sensors used in soft robots measure an electrical signal (e.g., electrical resistance, capacitance). The use of electrical sensors impedes progress toward electronics-free systems and adds complexity as separate control and power systems are required for sensing and actuation. A recent work has created a pneumatic strain

gauge for electronics-free soft robots, however, their sensitivity was low and narrow (i.e., ~0.7-1) [R14]. Therefore, we believe the sensitivity tunability of the STV holds significance, especially considering that programmable sensitivity coupled directly with proportional control capabilities has not been previously achieved.

We revised the “Pressure Curve Profile Programming” section in page 7 as follows:

“This macroscopic buckling when $n = 4$ relaxes much higher normalized strain energy than when $n = 2$ (Supplementary Fig. 4b), producing the highest-slope pressure curve similar to an on/off switch where the chamber pressure P_{ch} is either 0 or P_s ^{4, 50}. We note that in terms of sensitivity, defined as $S = \delta(P_{ch}/P_s)/\delta\epsilon$, a 14-fold tunability is achieved ranging from 3.34 when $n = 1$ to 40.66 when $n = 4$ (See Supplementary Fig. 4c and Methods section ‘soft tensile valve characterization’).”

Also, to provide the sensitivity data, we revised the legend in Supplementary Fig. 4c in page 10 of “Supplementary information” file as follows:

Supplementary Fig. 4. Supplementary analysis results of the STV. ... c Normalized chamber pressure P_{ch}/P_s plotted against the extension length ΔL for different numbers of WHYs n ; $n = 1$ ($\epsilon_{max} = 0.46$, $S = 3.34$), $n = 2$ ($\epsilon_{max} = 0.32$, $S = 17.26$), $n = 3$ ($\epsilon_{max} = 0.27$, $S = 7.71$), and $n = 4$ ($\epsilon_{max} = 0.21$, $S = 40.66$). ...

Finally, to provide the method, we revised the “Soft Tensile Valve Characterization” section in page 13 as follows:

“We repeated this process five times and then averaged the results. The Sensitivity $S = \delta(P_{ch}/P_s)/\delta\epsilon$ was calculated at the intermediate of the maximum P_{ch}/P_s ($P_{ch}/P_s = 0.37$ for $n = 1$ and $P_{ch}/P_s = 0.5$ for $n = 2-4$).”

[R13] Yan, W., Fuh, HR., Lv, Y. *et al.* Giant gauge factor of Van der Waals material based strain sensors. *Nature Communications* **12**, 2018 (2021).

[R14] A. Koivikko *et al.*, Integrated stretchable pneumatic strain gauges for electronics-free soft robots. *Communications Engineering* **1**, (2022).

Comment 2-5: I find the grasping results in Fig 3 lack some rigour and quantitative representation; what was the success rate, how did the sensitivity help/assist the task and how was it chosen for the task.

Response 2-5: The authors thank the reviewer's valuable comments. In Fig. 3, we intended to focus on the values that STV can bring to a conventional soft gripper system. Soft grippers with soft actuators can adapt and passively conform to various surfaces, allowing higher grasping success rate of an arbitrary object compared to their rigid counterparts. However, the grasping success rate is still a factor highly dependent on various aspects of the soft actuators (e.g., number, orientation, material, and geometry), as well as the personal capabilities of the users. In this regard, we have demonstrated quantitative representations of features that can be brought about solely by incorporating the STV: continuous controllability in the holding force and the sensitivity tunability of the gripper. However, we tested the success rate with 6 fresh users and found that most of the users successfully grasped various objects mostly in their first tries.

For the gripping task in Fig. 3f, the sensitivity was tuned considering the weight range and fragility of the selected objects. (i.e., lower sensitivity mode could not grasp heavier objects and higher sensitivity mode could not grasp fragile lightweight objects without breaking). Therefore, the sensitivity could be selected and tuned to shift the range of graspable objects or perform more task-specific applications. For example, a lower sensitivity mode could be applied for in vivo manipulations that require high displacement precision, and a higher sensitivity mode could be applied for rough power-lifting tasks that require high maximum holding force.

Comment 2-6: The electronics-free soft gripper highlights the ability to regulate the inlet and outlet valve, and the tuning seems to allow grasping of delicate and high force objects. I find it hard to see the significant advance of this application in comparison to other control methods for soft actuator control, as this could essentially also be achieved by a human triggering a valve, or pulling on tendons.

Response 2-6: The authors thank the reviewer's comments. A human triggering a soft valve has only been explored in controlling the inlet channel resistance with fixed outlet channel resistance. Using a fixed outlet channel resistance faces a trade-off between flow leakage and switching time; a small outflow resistance increases loss of air which could further make pump systems to fail in reaching target pressure state if the resistance is too small, whereas a large outflow resistance slows down the transition of the output pressures from high to low [R15]. The STV controls the pneumatic resistance of both inlet and outlet channels in a programmed manner with a counter mechanism overcoming this trade-off (when inlet channel resistance decreases, the outlet channel resistance simultaneously increases and vice versa). This counter resistance control mechanism would be difficult to achieve in a compact form factor similar to the STV with simple human triggering a valve. Also, a separate triggering of both inlet and outlet valves would delay real-time response of soft robots.

A human pulling on tendons directly transmits human force to the actuators. Therefore, a high force actuation can only be achieved by a human transmitting a high force, which could limit applications such as high force repetitive actuations. Also, handling highly delicate objects require additional spring mechanisms and set-ups to attenuate the motion response of the tendon-driven actuators. To overcome these drawbacks and tune the sensitivity of the actuators, tendon-driven systems require electronic components to measure inputs from humans and convert them to leveraged values. On the other hand, the STV can program the gripper's sensitivity independently of the pulling force for the handling of high force or delicate objects by adjusting the supply pressure without requiring additional set-up processes or electronics, which is advantageous for applications with unstructured environments. We revised the “Untethered and electronics-free soft gripper” section in page 9 as follows:

“In addition, by changing the constant supply pressure P_s through simple readjustment of the mechanical regulator, we show in Fig. 3e that different modes of our soft gripper can be programmed between two extremes with high maximum holding force with high sensitivity or high motional resolution with low sensitivity using the same STV (thus, independently of the STV pulling force).”

[R15] A. Rajappan *et al.*, Logic-enabled textiles. *Proc. Natl. Acad. Sci. USA* **119**, e2202118119 (2022).

Comment 2-7: I am unsure as to the motivation behind the CFD simulations. The pneumatic systems is relatively simple (i.e. in terms of the volume/flow rate), are their any more complex flow interactions (vorticies) or other behaviours the authors expected to motivate CFD analysis?

Response 2-7: We appreciate the reviewer's valuable comment. The CFD simulations were conducted to verify and predict our experimental results. The pressure drop in tubes with perfect circular cross-sections can be solved analytically using the equation $\Delta p = 128\mu Lm/\pi\rho R^4$ [R16], where m is the mass flow rate of air, μ is the dynamic viscosity of air, ρ is the density of air at standard pressure and temperature, and D and L are the inner diameter and the length of the tubing. However, the exact solution for Poiseuille flow through the deformed channel geometries is not explicitly known [R17], and the STV creates non-circular cross-sections and spiral geometry that continuously change under extension, making analytical solutions non-trivial. As a result, we were motivated to use numerical approaches rather than analytical approaches. In addition, exploiting the numerical CFD simulation could further give us insights, for example, how pressure is distributed inside the tubes (Supplementary Fig. 2). We revised the “Computational fluid dynamics” section in page 15 as follows:

“Since the analytical solution to the pressure drop in a deformed channel is not trivial, we coupled our finite element mechanical simulations with computational fluid dynamics (CFD)”

simulations to predict the output chamber pressure curves and aid future design.”

[R16] W.-K. Lee *et al.*, A buckling-sheet ring oscillator for electronics-free, multimodal locomotion. *Science Robotics* **7**, eabg5812 (2022).

[R17] A. Konda *et al.*, Reversible Mechanical Deformations of Soft Microchannel Networks for Sensing in Soft Robotic Systems. *Advanced Intelligent Systems* **1**, (2019).

Response to reviewer #3's comments

General Comment: In this manuscript, the authors describe a soft pneumatic valve that produces an analog pressure output in response to tensile strain. This strain-sensing valve, termed an STV by the authors, is constructed from a pair of concentric elastomeric tubes that act as flow conduits. A helically wrapped inextensible yarn pinches the inner tube and generates a space between the inner and outer tubes when stretched, and the corresponding changes in flow resistance enable the STV to behave as a fluidic voltage divider, producing an output pressure that changes in response to applied strain. The authors demonstrate the use of their device in controlling a soft gripper and an elbow assist device, both of which respond to mechanical strain applied to the STV. The manuscript is interesting and represents a timely

contribution to the ongoing effort in soft robotics toward developing soft sensors and transducers that directly output fluidic pressure signals.

However, additional details on the performance and characteristics of the valve, as outlined in the subsequent comments, must be included to provide a more complete picture of its capabilities and limitations. This expansion of detail will considerably aid in customization and deployment of the authors' approach in future soft robots and wearable devices. Furthermore, there has been considerable research interest in developing soft sensors—including pneumatic strain sensors—with several papers published in recent years; the advantages and limitations of the authors' presented device needs to be more clearly articulated in comparison with these earlier approaches. Specific questions and comments for the authors are listed below.

Response for General Comment: We appreciate the reviewer's profound and valuable comments for our manuscript. All comments are highly helpful and valuable in improving our paper. We modified our manuscript as best we could based on the reviewer's comments and addressed each comment one by one as follows.

Comment 3-1: 1. The reference list has omitted several related relevant works. The authors are encouraged to cite recent literature on fluidic sensors and supporting soft and fluidic control infrastructure relevant to their work. A few examples, mostly from 2022, are listed below (this list is not exhaustive; the authors are encouraged to search for additional relevant work):

[1] Hevia, et al., "High-Gain Microfluidic Amplifiers: The Bridge between Microfluidic Controllers and Fluidic Soft Actuators," *Advanced Intelligent Systems* (2022).

[2] Davletshin, et al., "A Bidirectional Soft Diode for Artificial Systems," *Advanced Functional Materials*, 2200658 (2022).

[3] Rajappan, et al., "Logic-Enabled Textiles," *Proceedings of the National Academy of Sciences*, 119, 35 (2022).

[4] van Laake, et al., "A Fluidic Relaxation Oscillator for Reprogrammable Sequential Actuation in Soft Robots," *Matter*, 5, 9 (2022).

[5] Decker et al., "Programmable Soft Valves for Digital and Analog Control," *Proceedings of the National Academy of Sciences*, 119, 40 (2022).

[6] Koivikko et al., "Integrated Stretchable Pneumatic Strain Gauges for Electronics-Free Soft Robots," *Communications Engineering* 1, 14 (2022).

[7] Jin, et al., "Mechanical Valves for On-Board Flow Control of Inflatable Robots," *Advanced Science*, 8, 2101941 (2021).

[8] Song, et al., “CMOS-Inspired Complementary Fluidic Circuits for Soft Robots,” *Advanced Science*, 8, 2100924 (2021).

[9] Souri et al., “Wearable and Stretchable Strain Sensors: Materials, Sensing Mechanisms, and Applications,” *Advanced Intelligent Systems*, 2, 2000039 (2020).

[10] Yeo et al., “Flexible and Stretchable Strain Sensing Actuator for Wearable Soft Robotic Applications,” *Advanced Materials Technologies*, 1, 1600018 (2016).

[11] Mengüç, et al., “Wearable soft sensing suit for human gait measurement,” *The International Journal of Robotics Research*, 33, 1748 (2014).

Response 3-1: We thank the reviewer’s excellent suggestion. We updated the reference list with suggested works and additionally searched recent relevant works in the “Introduction” section in page 2 and 3 as follows:

“The actuation of soft robots has been demonstrated by pneumatic¹¹⁻¹⁶”

...

“As a step towards this goal, recent efforts have created soft analogues of individual components, including electronic skins and soft strain sensors capable of self-sensing for monitoring robot and human motions^{10, 33-39}, and soft valves that can create nonmonotonous motions⁴⁰⁻⁵¹ and feedback control^{4, 47, 50, 52, 53}.”

The additionally searched recent relevant works are listed below.

[41] C. Qiao, L. Liu, D. Pasini, Bi-Shell Valve for Fast Actuation of Soft Pneumatic Actuators via Shell Snapping Interaction. *Advanced Science* 8, (2021).

[43] F. Zhao et al., Rapid, Energy-saving Bioinspired Soft Switching Valve Embedded in Snapping Membrane Actuator. *Journal of Bionic Engineering* 20, 225-236 (2022).

Comment 3-2: 2. Regarding the omitted references mentioned in the previous comment, reference [5] demonstrates an analog pressure regulator that produces an output pressure proportional to force input from a human user. If the system described in this work were attached to a rubber band to convert strain to applied force, it seems it would generate essentially the same output as the authors’ submitted work. The distinction from prior work or advantages of the present work should be emphasized.

Response 3-2: We thank the reviewer’s incisive comment. The reference paper [5] (ref. [52] in the manuscript) cleverly exploits the kinking of soft tubes to create a valve with the capability of on/off digital control as well as continuous analog control with some adjustments in design. We made great effort to thoroughly understand the ref. 52, including contacting the first author of this paper. There are several distinctions of the STV we would like to point out compared to

the valve in this paper.

1. Simple and compact: The STV demonstrates continuous control capability in a simple and compact manner. The analog control demonstration in the reference paper [5] required separate preparation and connections of 60 cm tubes to each pneumatic finger actuator to release air from the actuators to the atmospheric pressure. This could unwillingly require an additional process for different actuator control, as well as limit the compactness and portability of the system for untethered outdoor operations and applications with large numbers of actuators. On the other hand, due to the helical pinching mechanism, the STV seamlessly integrates the inlet and outlet tubes in a core-shell structural manner allowing operations with a simple one-step connection to actuators without requiring auxiliary parts or processes. Furthermore, the diameter of the STV is as low as a single silicone tube (5 mm) which we expect to be difficult to achieve using the mechanisms and assemblies demonstrated in the reference paper [5] (~15 mm) while maintaining the valve performance.
2. Effective method to simultaneously modulate inlet and outlet resistance: Using a fixed outlet channel resistance faces a trade-off between flow leakage and switching time; a small outflow resistance increases loss of air which could further make pump systems to fail in reaching target pressure state if the resistance is too small, whereas a large outflow resistance slows down the transition of the output pressures from high to low [R18]. Therefore, the outflow resistance must be within an acceptable range which also creates a loss of compressed gas at all output states. The STV with helical pinching mechanism overcomes this trade-off with the variable inlet and outlet resistance, yielding fast transitions of the output from high to low with no flow leakage at high and low outputs (please see **Response 3-5**).
3. High programmability within compact form: The STV is highly programmable while maintaining the form factor. The reference paper [5] demonstrated only a single output pressure vs. force curve using kinking of soft tubes for the inflow control. This kinking mechanism creates a fixed shape-morphing path (resulting in a fixed airflow resistance path) inside the channels during deformation and would require additional mechanisms or components for the path tuning. On the other hand, the STV exploits helical pinching of the soft tubes which allows tuning of cross-sectional shape paths without increasing the structural complexity. Therefore, various output pressure curves can be achieved within the same compact form factor.
4. Fully soft and flexible: In the reference paper [5], thermoplastic polypropylene straws are used to position and derive kinking of the soft silicone tubes. We expect these straws are semi-rigid which could potentially show an unsafe, rigid response in human-machine interactions with limitations in flexibility (e.g., bending capabilities). On the other hand, the STV exploits entirely of rubbery materials, making it more fracture-resistant, flexible, safe under human-machine interactions, and advantageous for potential integration into entirely soft systems (Please refer to **Response 3-8**).

We revised the “introduction” section in page 3 as follows:

“However, soft sensors alone still need other electronic components for control, while most state-of-the-art soft valves fundamentally operate as on/off switches, which limits their perceptive capabilities to interact with continuously changing environments.”

Also, we added a table comparing our work with relevant recent literature in Supplementary Table 1 (please refer to **Response 3-3**).

[R18] A. Rajappan *et al.*, Logic-enabled textiles. *Proc. Natl. Acad. Sci. USA* **119**, e2202118119 (2022).

Comment 3-3: 3. Reference [6] above describes a pneumatic strain sensor that also uses a fluidic resistance based voltage divider, similar to the approach taken in this work. Again, the distinction from prior work or advantages of the present work should be emphasized.

Response 3-3: We thank the reviewer’s comment. The reference paper [6] (ref. [36] in the manuscript) presents pneumatic strain sensors based on meandering microchannels inside a soft silicone material, which create relative change in pneumatic resistance under stretching or compression of the sensor. Previous works [R19-22] including the reference paper [6] have demonstrated actuation of soft robots using microfluidic devices, however, these devices create low flow rates for practical robot applications where macroscopic motions and forces are required [R23]. In the reference paper [6], the strain sensor was parallelly connected to pneumatic grippers to act as a pneumatic valve. However, the gripper took ~8 s to ~1 min just to close under applying a substantial compressive force to the sensor. Besides the fact that the reference paper [6] has not demonstrated the analog control capability using the soft sensors, the actuation speed in this range is very slow for practical robotic applications (e.g., power-gripping or human-robot interactions demonstrated in our work). Furthermore, the control system in the reference paper [6] has fixed inflow resistance (contrary to the reference paper [5] which has fixed outflow pneumatic resistance, see **Response 3-2 part 2**) which also faces a trade-off between switching time and flow leakage: a small inflow resistance increases loss of compressed gas when the output is low, whereas a large inflow resistance slows down the transition of the output from low to high. For the better understanding of our work’s position, we added a table comparing our work with relevant recent literature in Supplementary Table 1 in page 19 of “Supplementary information” file as follows:

Capability	Our valves	Micro-Fluidics ^{23, 40}	Bistable silicone valves ^{4, 49, 50, 53}	Digital, analog combinative valves ⁵²	Bidirectional check valves ⁴¹	Sheet-based valves ⁴⁷	Hysteretic valves ⁴⁶	Stretchable strain gauge ³⁶
Mechanism	Helical pinching	Linear pinching	Kinking	Kinking	Leaflet Flapping	Kinking	Slits in soft membrane	Channel elongation
Device shape	Tubular	Rectangular/Sheet	Cylindrical	Tubular	Tubular	Sheet	Tubular	Rectangular
Characteristic scale	Diameter = 5 mm	Thickness = 1 ~ 2.8 mm	Diameter = 27 mm, height = 34 mm	Diameter = 15 mm*	Channel length = 4 ~ 8 mm	Thickness = 2 mm**	Diameter = 5 mm	92 x 15 x 4 (mm)***
Practical flow rates for actuation	✓	—	✓	✓	✓	✓	✓	—
proportional control capabilities	✓	—	—	✓	—	—	—	—
Programmable proportional control	✓	—	—	—	—	—	—	—
Operation with single, constant pressure source	✓	—	✓	✓	—	—	✓	—
Proportional self-sensing capabilities	✓	—	—	✓	—	—	—	✓
Inflow and outflow resistance control	✓	—	—	—	—	—	—	—
Entirely soft	✓	✓	✓	—	✓	✓	✓	✓

* Estimated from Fig. 1c in⁵²
**Estimated from Fig. S13 in⁴⁷
*** Estimated from Fig. S13 in³⁶

Supplementary Table 1. Comparison of our valves to recently developed soft control devices.

(We note the references in this table can be found in page 17-19 of the “Reference” section of the manuscript.)

Also, we revised the sentences in “Introduction” section in page 4 as follows:

“This single mechanism enables simultaneous and reciprocal transformations of both inlet and outlet channels, allowing the STV to be designed in a compact, seamlessly integrated linear form (diameter = 5 mm) advantageous for embodiment into soft systems and human-robot interactions. A detailed comparison of the STV with state-of-the-art is shown in Supplementary Table 1.”

[R19] Wehner, M. *et al.* An integrated design and fabrication strategy for entirely soft, autonomous robots. *Nature* **536**, 451-455, doi:10.1038/nature19100 (2016).

[R20] Q. D. Zhang, M. Zhang, L. Djeghlaf, J. Bataille, J. Gamby, A. M. Haghiri-Gosnet, A. Pallandre, Logic digital fluidic in miniaturized functional devices: Perspective to the next generation of microfluidic lab-on-chips. *Electrophoresis* **38**, 953–976 (2017).

- [R21] P. N. Duncan, T. V. Nguyen, E. E. Hui, Pneumatic oscillator circuits for timing and control of integrated microfluidics. *Proc. Natl. Acad. Sci. USA* **110**, 18104–18109 (2013).
- [R22] P. N. Duncan, S. Ahrar, E. E. Hui, Scaling of pneumatic digital logic circuits. *Lab Chip* **15**, 1360–1365 (2015).
- [R23] D. J. Preston *et al.*, A soft ring oscillator. *Science Robotics* **4**, eaaw5496 (2019).

Comment 3-4: 4. In the abstract, the authors state that the STV “integrates the functional capabilities of sensors, controllers, and pneumatic valves” (lines 30–31) and realizes “physical sharing of both sensing and control structures” (line 33). It is not clear if the current device can be regarded as incorporating any “control” capabilities, as it functions only as a strain sensor, i.e., a transducer that converts a mechanical input signal to a pressure output signal. For example, a strain bridge (which acts as an electronic voltage divider that also responds to strain) may be used in conjunction with a voltage source to provide an analog voltage output that could, in principle, drive an electrical actuator; however, it is still regarded only as a sensor, and not as an analog controller. Therefore, the aforementioned claim should either be moderated or properly justified.

Response 3-4: We thank the reviewer’s excellent suggestion. We agree that the statement regarding our device incorporating the capabilities of controllers can be highly controversial. The original intention was to refer to the functional capability of an embedded controller that is widely used in pre-programmed electronic devices for specific applications. While the STV cannot be *in situ* programmed to yield different output pressure curves, this function bridges the capabilities of sensors and pneumatic valves, controlling the output pressures in a preprogrammed manner according to the strain. However, traditional controllers, strictly, have straight accessibility to measured variables from sensors and operate on the output with real-time programmability. Therefore, we tone down the suggested statements by using the term “control valves”. The control valves are not regarded as controllers yet have the capability to control fluidic flow rates or pressures (contrary to the on/off valves, which either enable unrestricted flow or entirely shut off the flow).

we revised the sentences in the “Abstract” section in page 2 as follows:

“Here, we report a soft self-sensing tensile valve that integrates the functional capabilities of sensors and control valves to directly transform applied tensile strain into distinctive stable output pressure states using only a single, constant pressure source.”

...

“By harnessing a unique mechanism, “helical pinching”, we derive physical sharing of both sensing and control valve structures, achieving all-in-one integration in a compact form factor.”

Comment 3-5: 5. At intermediate output pressures (when the inlet and outlet channels are partially open), the STV incurs a continuous loss of pressurized gas from the supply source to the atmosphere, which could pose a serious limitation when using finite and exhaustible onboard gas sources (such as a gas canister). Could the authors quantify the gas loss through their device as a function of the level of strain, and also comment on the longevity of their portable demos?

Response 3-5: The authors thank the reviewer’s incisive comments. As the reviewer mentioned, the STV incurs a continuous loss of pressurized gas at intermediate output pressures. We redid the experiment and measured the flow rate of gas exhaust and found a maximum flow rate of 1.81 LPM and an average flow rate of 0.81 LPM over the strain range. We calculated the gas volume of the CO₂ canister at standard temperature and atmospheric pressure $V = 95\text{g} * 0.5476 \text{ L/g} = 52 \text{ L}$, with the density value obtained from the MSDS. Therefore, the portable demo last roughly 29 minutes (52/1.81) if the STV is maintained at the highest gas loss and 64 minutes (52/0.81) on average. Such longevity could be increased by using multiple or higher-capacity CO₂ canisters or/and by reducing the exhaust flow rate of the STV. We connected an inlet resistor (inner diameter of 0.4 mm and length of 20 mm) to the STV and reduced the maximum and average flow rates to 1.13 LPM and 0.51 LPM respectively. This result in increased longevity of 46 minutes at a maximum gas loss and 100 minutes on average. We added this information in Supplementary Fig. 5b in page 11 of “Supplementary information” file as follows:

Supplementary Fig. 5. Reliability, durability, and repairability of the STV. ... b (i) Flow rate Q as a function of the normalized strain ϵ/ϵ_{max} ($Q_{max} = 1.81 \text{ LPM}$, $Q_{average} = 0.81 \text{ LPM}$). The inset represents inlet resistor with inner diameter of 0.4 mm and length of 20 mm connected to the inlet of the STV, reducing the flow rate to $Q_{max} = 1.13 \text{ LPM}$, $Q_{average} = 0.51 \text{ LPM}$ ($n = 2$, $p = 10 \text{ mm}$, $L_0 = 100 \text{ mm}$). (ii) Duration of the portable demonstrations calculated with the highest and average flow rate Q

Also, we added a discussion in Supplementary Note 3.2 in page 5 and 6 of “Supplementary information” file as follows:

3.2 Longevity of the portable demo

“The STV incurs a continuous loss of pressurized gas at intermediate output pressures (Supplementary Fig. 5b). We measured the flow rate of gas exhaust and found a maximum flow rate of 1.81 LPM and an average flow rate of 0.81 LPM over the strain range. We calculated the gas volume of the CO₂ canister at standard temperature and atmospheric pressure $V = 52 \text{ L}$, with the density value obtained from the MSDS. Therefore, the portable demo last roughly 29 minutes if the STV is maintained at the highest gas loss and 64 minutes on average. The

longevity could be increased by using multiple or higher capacity CO₂ canisters or/and by reducing the exhaust flow rate of the STV. We connected an inlet resistor (inner diameter of 0.4 mm and length of 20 mm) to the STV and reduced the maximum and average flow rates to 1.13 LPM and 0.51 LPM respectively. This result in increased longevity of 46 minutes at the maximum gas loss and 100 minutes on average.”

Comment 3-6: 6. Neither the finite element analysis nor the analytical model presented in the manuscript seems to account for loads arising from differential gas pressure across the tube walls, and no justification is provided for their exclusion. Do forces arising from internal gas pressure inflate the tubes or otherwise influence their deformation profile? Does varying the supply pressure change the input-output response of the STV? It might be that pressure forces can be neglected due to the small diameter of the tubes used; if this is indeed the case, can the authors provide a suitable quantitative scaling to show that the pressure forces are much smaller than the contact forces exerted by the yarns?

Response 3-6: The authors thank the reviewer’s valuable comments. We did run simulations and developed analytical models considering the deformation arising from the gas pressure across the tube walls. We apologize for not properly explaining them. Before developing CFD and analytical models, we tested how the diameters of the inner and outer tubes change under different supply pressures (Supplementary Fig. 10e). In the case of the inner tube, since inner tube had a small inner diameter of 2 mm, a thickness of 1 mm, and the yarn wrapping around the outside of the inner tube, the change according to the supply pressure was very small ($< 70 \mu\text{m}$ at $P_s = 100 \text{ kPa}$) and slightly increased when $P_s = 120 \text{ kPa}$ ($\sim 380 \mu\text{m}$). However, in the case of the outer tube, which had a larger inner diameter of 4 mm and a smaller thickness of 0.5 mm, the diameter slightly increased and then began to drastically increase near when $P_s = 120 \text{ kPa}$.

For the analytical models (Supplementary Fig. 7 and 8), since the load created by the supply pressure had a very small effect on the inner tube (not significant enough to deform the channel geometry), we neglected the load arising from the differential gas pressure. For the CFD simulation, since the outer tube had slight diameter expansions, we accordingly scaled up the outer tube after when we extracted the deformed geometry from the previous FEA extension simulations (by a factor of 1.022 at 60 kPa). We added the data in Supplementary Fig. 10e in page 16 of “Supplementary information” file as follows:

Supplementary Fig. 10. Mechanical characterizations. ... e Outer diameter of the inner and outer tube according to the supply pressure P_s .

Also, we revised the paragraph in Supplementary note 1 in page 3 of “Supplementary information” file as follows:

“Thus, for $n = 1$, which causes translational movement of the inner tube, the solution can be obtained by replacing equation (S1) with the translational loading pattern in which $\bar{P} = \bar{P}_0(c_1 + c_2 \cos(2\tilde{\theta}) + c_3 \cos(3\tilde{\theta}) + c_4 \cos(4\tilde{\theta}) + c_5 \cos(5\tilde{\theta}) + c_6 \cos(6\tilde{\theta}))$. Also, while the load arising from the supply pressure could influence the geometry the inner tube, the load had negligible effect up to $P_s = 120$ kPa (see Supplementary Fig. 10e).”

Also, we revised the paragraph in “Computational fluid dynamics” section in page 15 as follows:

“We exported the morphed surface data at multiple extension points from the FEA results (see Methods section ‘Finite element simulations’) and then reconstructed it with a retopology algorithm quad-remesher (Rhinoceros 7) to create manageable polygon meshes from the existing surfaces. We accounted for the load caused by the supply pressure, which expanded the outer tube (the inner tube had negligible effect), by 3D scaling up the deformed outer tube geometry proportionately (by a factor of 1.022 at 60 kPa, see Supplementary Fig. 10e). Then, we extracted the topology of the inlet and outlet channels and connected their ends with a plug using Boolean commands (Supplementary Fig. 2a).”

Regarding the effect of the supply pressure on the STV’s input-output response, we found out that the normalized pressure curve increased from low supply pressures, then converged as the supply pressure increased. This result might be due to the following reason:

Since the channels of the STV create non-circular cross-sections and spiral geometry that continuously change under extension, the exact analytical solution for Poiseuille flow through

the deformed channel geometries is not explicitly known [R24]. However, in order to understand the trend, we simplified the problem to a laminar flow in a perfect cylindrical pipe of radius D . Then. The inlet pneumatic resistance R_{in} can be roughly described to be inversely proportional to the fourth power of the characteristic diameter D from the Hagen-Poiseuille equation as follows:

$$R_{in} = 8\mu L / \pi D^4 \quad (R1)$$

Where μ is the dynamic viscosity of air, L is the length of the tubing, and D is the inner diameter of the tubing. Although the diameter of the outer tube slightly changed at $P_s = 20$ kPa ($\sim 10\mu\text{m}$) and 40 kPa ($\sim 50\mu\text{m}$), since the initial value of the R_{in} is extremely high ($\approx \infty$, as the inlet channel is initially closed), the R_{in} changed greatly according to Eq. (R1) and ultimately the P_{ch} increased as the P_s increased according to Eq. (1). However, from $P_s = 60$ kPa, despite the diameter of the outer tube changed relatively dramatically, since the initial R_{in} is small and Eq. (R1) rapidly saturates to an asymptote, R_{in} does not change significantly, and the P_{ch} graph converges. we added the Data in Supplementary Fig. 4h in page 10 of ‘‘Supplementary information’’ file as follows:

Supplementary Fig. 4. Supplementary analysis results of the STV. ... h Normalized chamber pressure P_{ch}/P_s as a function of normalized strain ϵ/ϵ_{max} for different Supply pressure P_s ($n = 2$, $p = 10$ mm, $L_0 = 100$ mm) ...

[R24] A. Konda *et al.*, Reversible Mechanical Deformations of Soft Microchannel Networks for Sensing in Soft Robotic Systems. *Advanced Intelligent Systems* **1**, (2019).

Comment 3-7: 7. A key characterization that is missing is the input force required to stretch the STV from zero to full extension, which can offset some of the assistive effort provided by the actuator in wearable assistive applications. There is currently only a passing mention of the magnitude of this force on line 227, where the authors state that it is of the order of 10 N, which seems nonnegligible. A force vs elongation curve for the STV should be straightforward to obtain using the authors’ stated experimental capabilities. Does this force curve change with the supply pressure?

Response 3-7: The authors thank the reviewer’s excellent suggestion. The force data was not

initially collected but rather we made rough estimates using weights during the development of the exosuit. However, now we redid the experiment and collected the force vs elongation curve. The force required for extension was slightly lesser with increasing supply pressure, and this could be due to the axial force created by the supply pressure, which aids the extension of the STV. We added the data in Supplementary Fig. 4i in page 10 of “Supplementary information” file as follows:

Supplementary Fig. 4. Supplementary analysis results of the STV. ... i Force plotted against the extension length ΔL for different supply pressures P_s ($n = 2$, $p = 10$ mm, $L_0 = 80$ mm).

We also revised the “Autonomous and Self-adaptive Soft Exosuit” section in page 10 as follows:

“We designed our STV taking into account the restoring force of the STV under extension (see Supplementary Fig. 4i), which can hinder the effective assistive torque generated by the soft elbow actuator.”

To reduce the force required to stretch the STV, the outer tube with the high elastic modulus (Supplementary Fig. 10b) could be replaced with lower modulus material, and an addition of a strain-limiting layer or tendons could potentially block the radial expansion of the outer tube (which becomes more vulnerable to radial expansion with lower modulus).

Comment 3-8: 8. Although designed to be operated primarily under tension, it is conceivable that the STV might experience occasional compressive loads in wearable applications. How does the device fare under compression, buckling, bending, or kinking during routine use?

Response 3-8: The authors thank the reviewer’s valuable comments. We redid the experiment and found negligible effects of bending, buckling, and kinking on the actuator. We also found negligible effects of compression when STV is unstrained, yet the actuator pressure increased than the target pressure when the STV was strained and compressed 2 mm from the offset. This could be due to the complete pinching of the inner tube while the outer tube had some flow space at the side. We have added the data in Supplementary Fig. 6 in page 12 of “Supplementary information” file as follows:

Supplementary Fig. 6. STV under different mechanical loads. Images of the STV under *a* Bending, *b* buckling and kinking, *c* compression without extension, and *d* compression with strain. The actuator pressure increased more than the target pressure only when the STV was strained and compressed 2 mm from the offset.

We also revised the “Autonomous and Self-adaptive Soft Exosuit” section in page 9 as follows:

“The STV with a soft, compact, and linear form can also provide conformable and safe human–machine interactions (see Supplementary Fig. 6).”

Comment 3-9: 9. Scalability, cost, and robustness are all key considerations for soft valves intended for wearable applications. Could the authors provide more information on the durability of their STVs, e.g., the maximum supply pressure or the number of stretch cycles they can withstand before failure? Can the valves be repaired if damaged in use? How costly are they to manufacture, and can the fabrication process be scaled up for mass-production?

Response 3-9: The authors appreciate the reviewer’s comments. We agree that scalability, cost, and robustness are all crucial factors when considering soft valves for wearable applications. We have replied to each reviewer’s questions point-by-point as follows.

Response 3-9.1: Maximum supply pressure or the number of stretch cycles before failure

We found out functional activeness of the STV up to $P_s = 140$ kPa (Supplementary Fig. 4h, please refer to **Response 3-6**). However, the outer tube exponentially inflated at certain threshold supply pressure P_s (Supplementary Fig. 10e, please refer to **Response 3-6**), which caused the outer tube to burst at $P_s = 157$ kPa. This inflation of the outer tube could be potentially blocked by adding a strain-limiting layer or tendons that exert repulsive forces on the radial expansion of the outer tube.

Also, we repeatedly stretched the STV and found reliable performance up to 10,000 cycles. However, the STV faced functional degradation at 20,000 cycles and then the outer tube was

detached from the connector around 25,000 cycles. We added the data in Supplementary Fig. 5d in page 11 of “Supplementary information” file as follows:

Supplementary Fig. 5 Considerable factors of the STV during practical use. ... d P_{ch}/P_s plotted against the normalized strain ϵ/ϵ_{max} after repeated extension of the STV ($n = 2$, $p = 10$ mm, $L_0 = 100$ mm, $P_s = 60$ kPa). ...

Also, we added a discussion in Supplementary Note 3.4 in page 6 of “Supplementary information” file as follows:

3.4 Maximum supply pressure and the number of stretch cycles before failure

“The STV is functionally active up to $P_s = 140$ kPa (Supplementary Fig. 4h). However, the outer tube, which has a relatively larger inner diameter, and smaller wall thickness compared to the inner tube, exponentially inflates at certain threshold supply pressure P_s (Supplementary Fig. 10e), which caused the outer tube to burst at $P_s = 157$ kPa. This inflation of the outer tube could be potentially blocked by adding a strain-limiting layer or tendons that exert repulsive forces on the radial expansion of the outer tube.

For the durability test, we repeatedly stretched the STV with the speed of 2 seconds per cycle and collected P_{ch}/P_s vs ϵ/ϵ_{max} data at various cycles (Supplementary Fig. 5d). The STV had reliable performance up to 10,000 cycles. However, the STV faced functional degradation at 20,000 cycles and then the outer tube was detached from the connector around 25,000 cycles.”

Response 3-9.2: Repair after damage.

The authors agreed that a sharp cut of the outer tube and detachment of the outer tube from the connectors are the two most vulnerable failures that STVs can face during use. We found out we could functionally repair the STV using a non-adhesive silicone tape for sharp cuts of the outer tube, and silicone adhesive Silpoxy for the detachment of the outer tube from the connector. We have added this information in Supplementary Fig. 5e in page 10 of “Supplementary information” file as follows:

Supplementary Fig. 5. Considerable factors of the STV during practical use. ... e Functional repairment of the STV after failure. (i) Wrapping of non-adhesive silicone tape to repair a sharp cut of the outer tube and (ii) coating of silicone adhesive Silpoxy to repair a detachment of the outer tube from the connector.

Also, we added a discussion in Supplementary Note 3.5 in page 6 of “Supplementary information” file as follows:

3.5 Repair after damage

“Sharp cuts in the outer tube and detachment of the outer tube from the connector are two vulnerable damages that STVs may encounter during use. The STV can be functionally repaired using a non-adhesive silicone tape (HanyangMSL) for the sharp cuts of the outer tube (Supplementary Fig. 5e(i)), and silicone adhesive Silpoxy (Smooth-On) for the detachment of the outer tube from the connector tube (Supplementary Fig. 5e(ii)).”

Response 3-9.3: Cost regarding fabrication of the STV.

The cost of materials used for the fabrication of an STV ($n = 2$, $p = 10$, $L_0 = 80$ mm) was approximately \$0.586 in USD. We have added the above information in detail in Supplementary Table. 3 in page 21 of “Supplementary information” file as follows:

Material	Supplier	Price	Amount	Unit	Producible valves	Cost/valve
Inner tube	GSJ	\$45.59	1000	g	870	0.052
Outer tube	Inner MED	\$4.53	5	m	82	0.055
Polyethylene Yarn	Mark	\$6.30	100	m	385	0.016
Tango rubber	Stratasys	\$1518.00	3600	g	6741	0.225
Agilus rubber	Stratasys	\$1628.00	3600	g	10112	0.161
Permabond 2050	Permabond	\$22.20	28.3	g	353	0.062
ClearFlex30	Smooth-On	\$82.00	880	g	5176	0.015
Total cost/valve:						\$0.586

Supplementary Table 3. Material price estimate (in USD) for the STV fabrication ($n = 2$, $p = 10$, $L_0 = 80$ mm).

Also, we revised the “Conclusion” section in page 11 as follows:

“Using only low-cost (see Supplementary Table 3), widely accessible all-soft materials and a simple fabrication process, the STV facilely replaces the former requirement of electronic infrastructures for perceptive robots, opening the avenue for proportional applications of soft machines including in environments where electronics may not be compatible, such as in the presence of spark ignition, dust, or radiation, as well as in vivo or underwater environments.”

Response 3-9.4: Mass-production

STV is still in the research stage, thus automative manufacturing for mass-production will require further research. The core processes are listed below.

The fabrication of the STV can be divided into 3 core processes: helically wrapping the yarn onto the inner tube, dip-coating to fix the position of the yarn, and assembling other parts (i.e., outer tube and connectors). Automating some of the fabrication processes could potentially enable the mass-production of the STV.

For example, for the helical wrapping of the yarn (which is the most laborious process), incorporating a widely used large-scale technique to wrap yarns onto cylindrical cores using

motors and extruders [R25], or circular braiding technology [R26] could allow large-scale fabrication of inner tubes with helically wrapped yarns. Also, while we used inks and stamps to mark the guideline for the yarns on the inner tube, this process could be removed when using large-scale manufacturing systems as the motors and extruders would provide uniform wrapping with controllable pitch.

For the dip coating process, incorporating commercially developed large-scale dip coating machines or different techniques such as R2R fabrication system could potentially automate the process for mass-production.

Lastly, while the assembling process is relatively trivial, the automation could be achieved with specialized robots.

[R25] Zhang G, Ghita OR, Lin C, Evans KE. Large-scale manufacturing of helical auxetic yarns using a novel semi-coextrusion process. *Textile Research Journal* **88**, 2590-2601 (2018).

[R26] Jiang N, Hu H. Auxetic yarn made with circular braiding technology. *Phys. Status Solidi Basic Res.* **256**, 1800168 (2019)

Comment 3-10: Other minor comments: 10. In Fig. 3g, the authors show their soft gripper handling a lit sparkler to demonstrate the advantages of its electronics-free construction and suitability for environments involving sparks (line 214). This is not a relevant demonstration, as electronic circuits are affected by electromagnetic interference caused by electrical sparks, not those produced by chemical combustion. A demonstration of the device operating near a spark gap, welding arc, or an automobile spark plug would be more appropriate in this context.

Response 3-10: The authors thank the reviewer's valuable suggestion. As the reviewer suggested, we redid the gripper demonstration near a welding arc. We have added the data in Fig. 3g in page 22 as follows:

Fig. 3. Untethered and electronics-free soft gripper. ... g Photographs of the soft gripper handling a pinecone near a welding arc, and h coral reef under water.

Comment 3-11: 11. Were the elongation vs output pressure curves in Fig. 2 measured during both elongation and retraction of the STV? Given the buckling-dominated behavior for $n = 2$ and 4 yarns, it is possible that the valve response curve might show discernible hysteresis.

Response 3-11: The authors thank the reviewer's excellent comments. The pressure curves were originally measured only at elongations. However, now we redid the experiment for both elongation and retraction. we found negligible hysteresis when $n = 1, 3$, and hysteresis when $n = 2, 4$. We found larger hysteresis when $n = 4$ than when $n = 2$. We have added the data in Supplementary Fig. 4a in page 9 of "Supplementary information" file as follows:

Supplementary Fig. 4. Supplementary analysis results of the STV. a Cyclic loading and unloading results of the STV shown in Fig. 2c. ...

Also, we revised the “Pressure curve profile programming” section in page 6 as follows:

“Fig. 2c and Supplementary Fig. 4a show these curves in normalized form (P_{ch}/P_s) as a function of the normalized axial tensile strain (ϵ/ϵ_{max}) applied to the STV.”

Comment 3-12: 12. On line 22, the formula for the flow rate is stated as $Q = 15\pi vd^2$, whereas it should correctly read $Q = (1/4)\pi vd^2$. This could be a simple typographical error, but the authors should confirm that the correct numerical coefficient was used in the actual calculations, as it could significantly affect the determination of the Reynolds number and, by extension, the flow regime.

Response 3-12: The reviewer is correct. We apologize for the confusion. This was a mistake made during forming the equation with inconsistent use of unit time: Q (m^3/s) = $15\pi v$ (m/min) d^2 . The highest Q we obtained was 1.81 LPM which yields the fastest velocity of 9.55 m/s. We obtained Mach number $M = v/c$, and Reynolds number $Re = \rho vd/\mu$ by substituting this value into these equations. We reconfirmed the numerical coefficients in the calculation: $c = 340.27$ m/s, $\rho = 1.184$ kg/ m^3 , $\mu = 1.849 \cdot 10^{-5}$ m²/s. We have revised the suggested statement in “Chamber Pressure” section in page 12 as follows:

“To characterize the dependence of the chamber pressure P_{ch} of the STV on the supply pressure P_s and the airflow resistance of the inlet channel R_{in} and the outlet channel R_{out} , we first calculated the air velocity v using $Q = (1/4)\pi vd^2$ where Q is airflow rate obtained from data, and d is the inner diameter of the inner tube.”

...

“We estimated incompressible laminar flow with $M \approx 0.028$ and $Re < 1223$, which is much smaller than the Mach number limit ($M \approx 0.3$) for compressibility⁵⁸ and the critical Reynolds number $Re \approx 2,300$ for the transition to turbulence⁵⁹.”

Comment 3-13: 13. The depiction of intervals with negative pressure (outward arrows) in

Extended Data Fig. 4 is inconsistent with the functional form shown in Eq. S1, which is always non-negative.

Response 3-13: The authors thank the reviewer’s meticulous comments. We have fixed the depiction of the schematic in Supplementary Fig. 7 in page 13 of “Supplementary information” file as follows:

Supplementary Fig. 7. Large deformation beam theory predicting inner tube deformations.

...

REVIEWER COMMENTS

Reviewer #2 (Remarks to the Author):

I would like to thank for the authors for their thorough review and adaption of the paper in light of the comments. I think the increasing focus on the programming the output and the inclusion into Figure 2 is an improvement.

I still find there are a few concerns regarding the work.

1. I still find there is a lack of clear statement defining the helical routing for soft valves and how this is different previous state of the art.
2. For the pressure curve profile programming, could the authors give more details of what and which parameters should be tuned, so this approach can be better leveraged by other researchers. There is a lot of information in Fig 2 c,e,f, but how can this be leveraged for others to design the pressure curves for their applications.
3. Clarity of quantitative nature of the demonstrations. Although this has been improved, these are still somehow qualitative and also quite hard to even understand what is being shown (e.g. Fig 3g). Why also are the results in Fig 4 shown on a Mannequin. Furthermore in the results section, when the two demonstrations are presented, more information on how the pressure curve profile was programmed from these applications would be very beneficial.
4. Some of the new figures in particular are somewhat confusing. There it lots of information, but not a huge amount of analysis. In addition, for example on Fig 2b, the scale is somewhat to read, and I am sure

Reviewer #3 (Remarks to the Author):

Second review of "A Soft, Self-Sensing Tensile Valve for Perceptive Soft Robots" (NCOMMS-22-37267A, revised manuscript) by Jun Kyu Choe, Junsoo Kim, Hyeonsoo Song, Joonbum Bae, and Jiyun Kim

This work describes the design and characterization of a soft pneumatic valve that produces an analog pressure output in response to tensile strain. We thank the authors for their detailed point-by-point responses to our comments on the original manuscript. The revisions satisfactorily address the questions raised in our previous review, and the new experimental data provided by the authors add to the clarity and comprehensiveness of the work and will facilitate the future customization and deployment of the device by researchers in the field. We are happy to recommend publication of the revised manuscript.

We appreciate the inclusion of relevant references to prior work, and the addition of supplementary table 1 comparing the authors' device to other pneumatic valves in the literature. We do, however,

encourage the authors to verify some of the entries in this table; for example, “Digital analog combination valves” (column 5) may be capable of programmable proportional control by changing the tension in the elastic band, and “Sheet-based valves” (column 7), while not capable of analog control, only require a single, constant-pressure source.

Reviewer #4 (Remarks to the Author):

This manuscript presents a soft and stretchable tensile valve that can regulate the output pressure as a function of the applied mechanical strain. The device is comprised of soft inner and outer tubes with an intermediate wrapping helical yarn where the air flow resistance is achieved by a helical pinching mechanism. The authors comprehensively studied the working principle, performance of soft valves, and their programmability through experiments, and computational and theoretical models. Finally, the potential use of self-regulating valves has been demonstrated in a soft gripper robot and a soft wearable exosuit.

Overall, the concept of measuring the applied strain electronic-free or regulating the pressure by strain is interesting. The full systematic characterisation of the device in this study is also important for its practical use. However, there are several issues that hinder the implementation of the device in electronic-free soft robotics. Most of these concerns were highlighted and questioned by other reviewers as well. Although the manuscript has been significantly revised, those fundamental problems with operation of the proposed self-sensing valve still remain. First, as demonstrated in Supplementary Fig. 5a, the performance of the valve is highly strain-rate dependent. It is very problematic, especially in wearable applications where strain rate varies greatly between steady-state to rates as large as 1 s^{-1} . Therefore, it is not clearly discussed how these valves can accurately function in such a dynamic condition. Second, the valve suffers from continuous loss of the pressurised gas, which is not desirable for many applications as they require frequent replacement of the CO₂ canister. Third, the authors argued that their proposed valve can be used for intelligent sensing, control, and perception. However, none of the application demonstrations focused on how the full cycle of sensing, control, and perception can be achieved. Undoubtedly, the proposed design offers some benefits towards electronic-free robotic control, but at the same time, it brings some other complexities and challenges to a soft robotic system. Considering the above and based on the comments from other reviewers, the reviewer is not convinced that the current manuscript should be accepted for publication in Nature Communication. Another minor comment below:

1- We found the simulation to be in reasonable agreement with the analytical model shown in Supplementary Fig. 6 and the cross-sectional micro-CT images shown in Fig. 2b. Supplementary Fig. 6 seems to be displaced during the revision, please double check.

2nd Response to reviewers' comments for the manuscript – “A Soft, Self-Sensing Tensile Valve for Perceptive Soft Robots”

Response to reviewer #2's comments

General Comment: I would like to thank for the authors for their thorough review and adaption of the paper in light of the comments. I think the increasing focus on the programming the output and the inclusion into Figure 2 is an improvement.

Response for General Comment: We sincerely appreciate the reviewer's recognition of our efforts and valuable comments that improved the overall quality of our revised manuscript.

Comment 2-1: I still find there are a few concerns regarding the work. 1. I still find there is a lack of clear statement defining the helical routing for soft valves and how this is different previous state of the art.

Response 2-1: The authors thank the reviewer's valuable comment. We apologize that there was lack of clear statement regarding the helical routing and Supplementary Table 2.

We have added information about the helical routing structures in “Conclusion” section in page 11 as follows:

“By harnessing the helical pinching mechanism, a material-based approach to control pressure gradients in fluidic circuits, we demonstrated the STV that simultaneously self-senses its strain and generates corresponding output chamber pressures from a single, constant supply pressure. Helical routing structures have been previously explored to create various stretchable devices, including soft sensors and artificial muscles (see Supplementary Table 2), due to their stretchable nature as the fibers can be straightened out under tensile strain. Here, we exploited helical routing of yarns over a soft tube to develop the pinching mechanism that can control pneumatic resistance of the tube channel under tensile strain in a continuous and programmable manner. This mechanism is well identified by our computational and analytical models, which can further support the design of the output pressure curves produced by the STV for optimization in specific applications.”

Also, we have added information in Supplementary Table 2 in page 22 of “Supplementary information” file as follows:

	This work	Ref. ⁶⁴	Ref. ⁶⁵	Ref. ⁶⁶	Ref. ⁶⁷	Ref. ⁶⁸	Ref. ⁶⁹	Ref. ⁷⁰
Materials	Polyethylene yarn/ rubber tube	Spandex fiber/ silicone elastomer	Nylon/ copper wire	Kevlar fiber/ silicone elastomer	Tendon/ dexterous manipulator	Polyamide fiber/ polyurethane elastomer	Cellulose acetate, PLGA, polyurethane Nanofiber	Gold nanowire- impregnated fiber/ fiber conductor
Type	Pneumatic valve	Sensor	Artificial muscle	Pneumatic actuator	Continuum manipulator	Auxetics	Micro tissue	Supercapacitor
Mechanism	Inflow & outflow pneumatic resistance tuning	Capacitive	Radial-axial thermal expansion anisotropy	Strain limitation	Strain limitation	Interactive normal forces	Coil opening	Coil opening
Function	Proportional self- sensing and control of pneumatic actuators	Strain sensing	High work capacity contraction	Twist motion	S-shape motion	Negative Poisson's ratio	High stretchability	High stretchability

Supplementary Table 2. Comparison of our work to other devices using helical routing structures. *Helical routing structures can be characterized by the helical arrangement of fibers or wires around a central axis. Helical tendon routing has been explored in the role of action such as capacitive strain sensing, strain limited motions, auxetic and highly stretchable shape-morphing. This work exploits helical routing for the first time in soft valves to create a compact valve with proportional self-sensing and control capabilities.*

Comment 2-2: For the pressure curve profile programming, could the authors give more details of what and which parameters should be tuned, so this approach can be better leveraged by other researchers. There is a lot of information in Fig 2 c,e,f, but how can this be leveraged for others to design the pressure curves for their applications.

Response 2-2: The authors thank the reviewer's valuable comment. We have created a new workflow figure that guide the selection of various STV parameters and setup conditions. This information can be found in Supplementary Fig. 5 in page 12 of "Supplementary information" file as follows:

Supplementary Fig. 5. Flowchart for selecting STV geometrical parameters and setup conditions.

Also, to give authors more details on which parameters are related to the resulting output, we added a discussion in Supplementary Note 3.1 in page 5 of “Supplementary information” file as follows:

“3.6 STV design and setup

The geometrical design parameters of the STV and the setup conditions can be selected as shown in (Supplementary Fig. 5) for the use of STV in various applications.

Before the fabrication of the STV, the number of WHY n can be selected to target specific P_{ch}/P_s vs ϵ/ϵ_{max} curve shape. Then, after selecting the initial length of the STV (L_0), the maximum strain ϵ_{max} can be determined by selecting pitch p which equivalently determines the maximum extension length of the STV (see Supplementary Note 2).

After the fabrication of the STV, the supply pressure P_s can be selected to target the maximum P_{ch} . Finally, the original/inverse STV connection mode can provide increasing/decreasing P_{ch}/P_s vs ϵ/ϵ_{max} curve for the analog control of soft pneumatic actuators. The P_s and connection mode can be reprogrammed after the fabrication of the STV.”

Comment 2-3: Clarity of quantitative nature of the demonstrations. Although this has been improved, these are still somehow qualitative and also quite hard to even understand what is

being shown (e.g. Fig 3g). Why also are the results in Fig 4 shown on a Mannequin. Furthermore in the results section, when the two demonstrations are presented, more information on how the pressure curve profile was programmed from these applications would be very beneficial.

Response 2-3: The authors thank the reviewer’s valuable comments.

As for the clarity of quantitative nature of demonstrations, we intended to focus on the effectiveness and capability of the STV in replacing the role of the traditional electronic controller and valves. Therefore, we have provided quantitative information related to features that are highly dependent on the STV: continuous controllability in the holding force and the sensitivity tunability, in case of the gripper. Other quantitative information related to the gripper such as gripping success rate highly depend on various aspects unrelated to the STV: (i) type of soft actuators (e.g., number, orientation, material, and geometry), (ii) type of the grasping object (e.g., shape, orientation, and material property), (iii) personal capability of users (e.g., level of familiarity, skillfulness, etc.), and (iv) environment (e.g., humidity, dustiness). For example, same gripper can highly vary in performance by the environment conditions depending on the object type (Fig. X1a), and the success rate of gripping the same object can greatly vary by the actuator types (Fig. X1b) [R1]. Therefore, we excluded other quantitative information that cannot accurately and effectively represent the STV.

Fig. X1. Gripping success rate of various objects. **a** success rate variance depending on environment conditions. **b** success rate variance depending on gripper type. **c** selection of objects from #1 to #12.

Similarly, as for the mannequin for the soft exosuit in Fig. 4, the reason we used a mannequin

robot is to measure and evaluate the sole effectiveness of the STV in replacing the role of the electronic controller and valves, as there are a lot of variables and uncertainties when it comes to human subjects such as size, weight, fine movements, skeletal structure, and individual differences, which can affect the results of our experiment. Therefore, by using a mannequin robot in a standardized environment, we were able to eliminate these variables and focus solely on the effect of the STV in the soft exosuit.

The main focus of this paper is to propose the STV and its capabilities. The demonstrations we showed are just few examples on how the STV can be applied to replace traditional electronic controller and valves. The soft gripper and exosuit have various design and environmental variables other than the design of the STV which can affect other quantitatively definable features. Since these variables will still highly affect such features even when using traditional electronics and not the STV, we have only provided quantitative data that can effectively represent the STV and controlled the experiment so that variables not related to the STV are minimally involved.

As for the clarity of the figures, we modified Fig. 3g and Fig. 3h in page 23 of the manuscript to improve the understandability of the figures, as follows:

Fig. 3. Untethered and electronics-free soft gripper. ... g Photographs of the soft gripper handling a pinecone near a welding arc (which can affect electronic circuits by electromagnetic interference), and h coral reef under water.

Also, to provide more information on how the STV was programmed for the soft gripper, we revised the “Untethered and electronics-free soft gripper” section in page 9 of the manuscript as follows:

“We note that the STV was designed to provide continuous control with intermediate sensitivity S (with $n = 2$), to be well fitted to the soft gripper body ($L_0 = 80$ mm) with high

maximum strain ε_{max} (with $p = 10$), and to increase gripper pressure according to strain (with original connection). However, by programming the STV with different design or setup conditions, we may achieve binary on/off control (with $n = 4$), a gripper with different strain sensitivity (by tuning ε_{max}), or a reverse action gripper (with inverse STV connection) for other specific tasks.”

Also, to provide more information on how the STV was programmed for the soft exosuit, we revised “autonomous and self-adaptive soft exosuit” section in page 10 of the manuscript as follows:

“The flexion of the elbow is detected as strain and transferred to the STV by the tendon. Similarly to the soft gripper demonstration, the STV with the parameters $n = 2$, $p = 10$ mm, and $L_0 = 80$ mm was used, yet we additionally took into account the restoring force of the STV under extension (see Supplementary Fig. 4i) for the soft exosuit, which can hinder the effective assistive torque generated by the soft elbow actuator. Specifically, we first connected the STV in inverse mode (Fig. 4a (iii)). Then, we tied the tendon to the chamber connector of the STV and the forearm cuff, passing through a routing ring in the upper arm cuff, as shown in Fig. 4b, so that the STV was fully strained ($\varepsilon = \varepsilon_{max}$) at (i) $\theta = 0^\circ$ and restored ($\varepsilon = 0$) at (ii) $\theta = 60^\circ$ as the distance between the cuffs decreased. In this way, we achieved chamber pressure as an increasing function of the elbow flexion angle θ , as shown in Fig. 4c, without incorporating the restoring force of the STV at high angles (see Supplementary Fig. 13)”

[R1] Terrile, S., Argüelles, M. & Barrientos, A. Comparison of Different Technologies for Soft Robotics Grippers. *Sensors* **21**, 3253 (2021).

Comment 2-4: Some of the new figures in particular are somewhat confusing. There it lots of information, but not a huge amount of analysis. In addition, for example on Fig 2b, the scale is somewhat to read, and I am sure

Response 2-4: The authors thank the reviewer’s comments. The question that reviewer asked seemed to be cut off, so we could only write the answer as we understood it; however, if the question had a different meaning, please ask again.

With regards to the figure mentioned by the reviewer, it seems that it is probably Supplementary Fig. 2b and not Fig. 2b, as we found that the pressure scale bar was not correctly marked in Supplementary Fig. 2b. We apologize for the confusion. We have modified the pressure scale bar in Supplementary Fig. 2b in page 9 of “Supplementary information” file as follows:

Supplementary Fig. 2. Computational fluid dynamics simulations. *a* Schematic illustration of the inlet and outlet channels of the STV. The dotted circle represents where the chamber pressure P_{ch} lies inside the structure. *b* Pressure profile results inside inlet and outlet channels with x-z view (top), x-z cross-sectional view (middle), and y-z cross-sectional view (bottom) at different normalized strains (i) $\epsilon/\epsilon_{max} = 0.06$, (ii) $\epsilon/\epsilon_{max} = 0.5$, and (iii) $\epsilon/\epsilon_{max} = 0.8$.

Also, we added scale bars in Supplementary Fig. 3 to increase the clarity of the figure. This is in page 10 of “Supplementary information” file as follows:

Supplementary Fig. 3. *FEA results with transversal cross-sectional views at the increasing number of n . a $n = 1$, b $n = 2$, c $n = 3$, and d $n = 4$. Scale bars = 2 mm.*

Also, we added clear statement regarding the inflow and outflow resistance control in the legend of Supplementary Table 1 in page 21 of “Supplementary information” file as follows:

Capability	Our valves	Micro-Fluidics ^{23, 40}	Bistable silicone valves ^{4, 49, 50, 53}	Digital, analog combinative valves ⁵²	Bidirectional check valves ⁴¹	Sheet-based valves ⁴⁷	Hysteretic valves ⁴⁶	Stretchable strain gauge ³⁶
Mechanism	Helical pinching	Linear pinching	Kinking	Kinking	Leaflet Flapping	Kinking	Slits in soft membrane	Channel elongation
Device shape	Tubular	Rectangular/Sheet	Cylindrical	Tubular	Tubular	Sheet	Tubular	Rectangular
Characteristic scale	Diameter = 5 mm	Thickness = 1 ~ 2.8 mm	Diameter = 27 mm, height = 34 mm	Diameter = 15 mm*	Channel length = 4 ~ 8 mm	Thickness = 2 mm**	Diameter = 5 mm	92 x 15 x 4 (mm)***
Practical flow rates for actuation	✓	—	✓	✓	✓	✓	✓	—
proportional control capabilities	✓	—	—	✓	—	—	—	—
Programmable proportional control	✓	—	—	▲	—	—	—	—
Operation with single, constant pressure source	✓	—	✓	✓	—	✓	✓	✓
Proportional self-sensing capabilities	✓	—	—	✓	—	—	—	✓
Inflow and outflow resistance control****	✓	—	—	—	—	—	—	—
Entirely soft	✓	✓	✓	—	✓	✓	✓	✓

* Estimated from Fig. 1c in⁵²

** Estimated from Fig. S13 in⁴⁷

*** Estimated from Fig. S13 in³⁶

▲ May be capable by changing the tension in the elastic band

Supplementary Table 1. Comparison of our valves to recently developed soft control devices.

**** The STV provide effective method to simultaneously modulate inflow and outflow resistance. Using a fixed outlet resistance R_{out} to achieve analog control faces a trade-off between flow leakage and switching time⁵². If R_{out} is too small, it can lead to a high loss of air and cause pump systems to fail to reach the desired pressure state. On the other hand, if R_{out} is too large, the transition of output pressures from high to low can be slowed down. Therefore, R_{out} had to be carefully set within an acceptable range, which also causes a loss of compressed gas at all output states. The STV with a helical pinching control both R_{in} and R_{out} in a programmable manner (when R_{in} increases, R_{out} decreases simultaneously and vice versa). This results in fast transitions of output from high to low with no flow leakage at high and low outputs.

Also, to provide more analysis and show that the strain-rate dependency of the STV is highly due to the design of soft actuators, we redid the experiment with reduced initial chamber volume of 27,615 mm³ and modified Supplementary Fig. 6a in page 13 of “Supplementary information” file as follows:

Supplementary Fig. 6. Considerable factors of the STV during practical use. **a** Strain rate $\dot{\epsilon}$ dependence of the STV. (i) schematic illustration of the STV test set-up (ii) P_{ch}/P_s plotted against the normalized strain ϵ/ϵ_{max} for different strain rates $\dot{\epsilon}$ with initial actuator chamber volume of $27,615 \text{ mm}^3$ and (iii) $39,919 \text{ mm}^3$. When $\epsilon/\epsilon_{max} = 1$ is reached, the extension of the STV was maintained to allow the pressure inside the actuator to saturate to the target pressure. Inset: enlarged view near $\epsilon/\epsilon_{max} = 1$. The saturation time was (ii) 0.11 s, (iii) 0.3 s for $\dot{\epsilon} = 0.0109$ and (ii) 0.28 s, (iii) 1 s for $\dot{\epsilon} = 0.0150$ ($n = 2$, $p = 10 \text{ mm}$, $L_0 = 100 \text{ mm}$). ...

Also, regarding the strain-rate dependency of the STV, we have added more information in Supplementary Note 3.2 in page 5 and 6 of “Supplementary information” file as follows:

“3.2 Strain rate dependence

The output chamber pressure is a strain rate-dependent function due to the intrinsic limitation in the speed of air and the inflatable characteristics of the soft pneumatic actuators. As shown in Supplementary Fig. 6a, extending the STV at faster speeds could result in reduced output chamber pressures if the air from the inlet does not have enough time to move into the actuators to achieve the target chamber pressure state. Likewise, contracting the STV at higher speeds could result in increased output chamber pressures if the air from the actuator does not have enough time to exhaust. Also, the changes in air volume required for a soft actuator to reach a specific pressure is majorly dependent on the actuator’s initial chamber volume, geometry, and material properties. Since larger changes in air volume require longer time to reach the target pressure, the strain-rate dependency of the STV can be reduced, for example, by using an actuator with smaller initial chamber volume (Supplementary Fig. 6a (ii), (iii)).

In our system, the time required for the pressure to saturate for strain rate of 0.0109 s^{-1} and 0.0150 s^{-1} at $\epsilon = \epsilon_{max}$ was 0.11 s and 0.28 s respectively for initial chamber volume = $27,615 \text{ mm}^3$ and 0.3 s and 1 s respectively for initial chamber volume = $39,919 \text{ mm}^3$.”

Also, to provide more analysis regarding the STV under different mechanical loads, we added more information in the inset of the Supplementary Fig. 7. This is in page 14 of “Supplementary information” file as follows:

Supplementary Fig. 7. STV under different mechanical loads. Images of the STV under **a** Bending, **b** buckling and kinking (insets: enlarged images of the STV), **c** compression without extension, and **d** compression with strain (insets: schematic illustrations of the STV cross-section). The actuator pressure increased more than the target pressure only when the STV was strained and compressed 2 mm from the offset.

Also, to provide more analysis regarding the STV under different mechanical loads, we added a discussion in Supplementary Note 3.7 in page 7 of “Supplementary information” file as follows:

3.7 STV under different mechanical loads

The STV can conform to various mechanical loads and provide safe human-machine interactions. The effect of bending, buckling, kinking, and compression of the STV is shown in Supplementary Fig. 7. The actuator pressure did not increase during bending, buckling, and kinking of the STV as the outer tube was only induced to wrinkle partially and the cross-section of the inner tube outer was negligibly deformed. In detail, the inlet channel resistance, which is initially very high, was not reduced to allow the inflow of air.

For the compression, the actuator pressure did not increase when STV was not strained. In detail, although the outlet resistance was significantly increased by the compression, the inlet channel resistance, which is initially very high, was not reduced to allow the inflow of air.

However, the actuator pressure increased more than the target pressure when the STV was strained and compressed by 2 mm. When the STV is strained, the WHYS deform the inner tube which reduce the inlet channel resistance and increase the outlet channel resistance. In this state, the compression significantly increased the pneumatic resistance of the outlet channel, yet the inlet channel resistance, which is not initially extremely high, was not significantly increased and allowed some flow space at the sides.

Response to reviewer #3's comments

General Comment: Second review of “A Soft, Self-Sensing Tensile Valve for Perceptive Soft Robots” (NCOMMS-22-37267A, revised manuscript) by Jun Kyu Choe, Junsoo Kim, Hyeonsoo Song, Joonbum Bae, and Jiyun Kim

This work describes the design and characterization of a soft pneumatic valve that produces an analog pressure output in response to tensile strain. We thank the authors for their detailed point-by-point responses to our comments on the original manuscript. The revisions satisfactorily address the questions raised in our previous review, and the new experimental data provided by the authors add to the clarity and comprehensiveness of the work and will facilitate the future customization and deployment of the device by researchers in the field. We are happy to recommend publication of the revised manuscript.

Response for General Comment: We sincerely appreciate the reviewer's recognition of our efforts and favorable evaluation of our manuscript. The reviewer's comments and constructive suggestions greatly helped us to improve the overall quality of our revised manuscript.

Comment 3-1: We appreciate the inclusion of relevant references to prior work, and the addition of supplementary table 1 comparing the authors' device to other pneumatic valves in the literature. We do, however, encourage the authors to verify some of the entries in this table; for example, “Digital analog combination valves” (column 5) may be capable of programmable proportional control by changing the tension in the elastic band, and “Sheet-based valves” (column 7), while not capable of analog control, only require a single, constant-pressure source.

Response 3-1: The authors thank the reviewer's comment. We have modified some entries in the Supplementary Table 1 including the valves that the reviewer mentioned in page 21 of “Supplementary information” file as follows:

Capability	Our valves	Micro-Fluidics ^{23, 40}	Bistable silicone valves ^{4, 49, 50, 53}	Digital, analog combinative valves ⁵²	Bidirectional check valves ⁴¹	Sheet-based valves ⁴⁷	Hysteretic valves ⁴⁶	Stretchable strain gauge ³⁶
Mechanism	Helical pinching	Linear pinching	Kinking	Kinking	Leaflet Flapping	Kinking	Slits in soft membrane	Channel elongation
Device shape	Tubular	Rectangular/Sheet	Cylindrical	Tubular	Tubular	Sheet	Tubular	Rectangular
Characteristic scale	Diameter = 5 mm	Thickness = 1 ~ 2.8 mm	Diameter = 27 mm, height = 34 mm	Diameter = 15 mm*	Channel length = 4 ~ 8 mm	Thickness = 2 mm**	Diameter = 5 mm	92 x 15 x 4 (mm)***
Practical flow rates for actuation	✓	—	✓	✓	✓	✓	✓	—
proportional control capabilities	✓	—	—	✓	—	—	—	—
Programmable proportional control	✓	—	—	▲	—	—	—	—
Operation with single, constant pressure source	✓	—	✓	✓	—	✓	✓	✓
Proportional self-sensing capabilities	✓	—	—	✓	—	—	—	✓
Inflow and outflow resistance control****	✓	—	—	—	—	—	—	—
Entirely soft	✓	✓	✓	—	✓	✓	✓	✓

* Estimated from Fig. 1c in⁵²

** Estimated from Fig. S13 in⁴⁷

*** Estimated from Fig. S13 in³⁶

▲ May be capable by changing the tension in the elastic band

Supplementary Table 1. Comparison of our valves to recently developed soft control devices.

...

Response to reviewer #4's comments

General Comment: This manuscript presents a soft and stretchable tensile valve that can regulate the output pressure as a function of the applied mechanical strain. The device is comprised of soft inner and outer tubes with an intermediate wrapping helical yarn where the air flow resistance is achieved by a helical pinching mechanism. The authors comprehensively studied the working principle, performance of soft valves, and their programmability through experiments, and computational and theoretical models. Finally, the potential use of self-regulating valves has been demonstrated in a soft gripper robot and a soft wearable exosuit.

Overall, the concept of measuring the applied strain electronic-free or regulating the pressure by strain is interesting. The full systematic characterisation of the device in this study is also important for its practical use. However, there are several issues that hinder the implementation of the device in electronic-free soft robotics. Most of these concerns were highlighted and questioned by other reviewers as well. Although the manuscript has been significantly revised, those fundamental problems with operation of the proposed self-sensing valve still remain.

Response for General Comment: We appreciate the reviewer's valuable comments. We have carefully addressed each comment with point-by-point replies as follows.

Comment 4-1: First, as demonstrated in Supplementary Fig. 5a, the performance of the valve is highly strain-rate dependent. It is very problematic, especially in wearable applications where strain rate varies greatly between steady-state to rates as large as 1 s^{-1} . Therefore, it is not clearly discussed how these valves can accurately function in such a dynamic condition.

Response 4-1: The authors thank the reviewer's comment.

The performance of the STV is strain rate dependent and in a dynamic situation with high STV strain rate, the actuator will have a large delay in reaching the target pressure.

However, this is only conditionally true as the high delay is mainly due to the characteristics of soft pneumatic actuators. Compared to rigid actuators, the soft pneumatic actuators are highly inflatable and therefore require a significant more volume ($\sim 20x$) of air than their initial chamber volume to reach a target pressure [R2]. Specifically, the changes in air volume required for a soft actuator to reach a specific pressure is highly dependent on the actuator's initial chamber volume, geometry, and material properties. Unless the air flow rate is highly limited such as in microfluidic channels, the change in volume is the major factor that effect on the actuation speed (larger changes in volume require longer time to reach the target pressure). For example, a different geometry in the soft actuator (with same initial volume) can reduce the change in volume from $\sim 20x$ to $\sim 4x$ its initial volume, which enormously reduce a full actuation time from 3.3 s to 130 ms ($\sim 25x$) [R2] (the faster actuator even has a high actuator tip force). Also, it has been previously shown that an actuator fabricated with a stiffer material can yield $\sim 8x$ pressure from a same change in the volume (Fig. X2) [R2], which means that a huge amount of time can also be saved to reach a certain pressure with a stiffer actuator.

Fig. X2. Effect of Elastomer stiffness. Pressure–volume hysteresis curves for actuators fabricated with **a)** an extensible layer made from Ecoflex 30 and an inextensible layer made from PDMS and **b)** an extensible layer made from Elastosil M4601 and inextensible layer made from a composite of Elastosil M4601 and paper fabric.

In the demonstrations, we focused on the effectiveness of the STV in replacing the role of the traditional electronic controllers and valves. The STV introduces a method to target actuator pressures between the atmospheric pressure P_{atm} and the supply pressure P_s with practical flow rates for macroscopic soft actuators (Supplementary Table 1). The targeting of a particular pressure state is instantly set by the STV depending on its strain, but the essential cause of the highly strain-rate dependent pressure curves is the non-optimized design of the soft actuator, with intrinsic limitation in the speed of air. Therefore, we believe that the strain-rate dependency can be greatly reduced in the future with an optimized actuator design for very dynamic situations.

The demonstrated actuator in Supplementary Fig. 6a (used to be Supplementary Fig. 5a) had initial chamber volume of $39,919 \text{ mm}^3$. To show that the strain-rate dependency of the STV can be significantly reduced by the design of the actuator, we redid the experiment with reduced initial chamber volume of $27,615 \text{ mm}^3$ and modified Supplementary Fig. 6a in page 13 of “Supplementary information” file as follows:

Supplementary Fig. 6. Considerable factors of the STV during practical use. a Strain rate $\dot{\epsilon}$ dependence of the STV. (i) schematic illustration of the STV test set-up (ii) P_{ch}/P_s plotted

against the normalized strain $\varepsilon/\varepsilon_{\max}$ for different strain rates $\dot{\varepsilon}$ with initial actuator chamber volume of 27,615 mm³ and (iii) 39,919 mm³. When $\varepsilon/\varepsilon_{\max} = 1$ is reached, the extension of the STV was maintained to allow the pressure inside the actuator to saturate to the target pressure. Inset: enlarged view near $\varepsilon/\varepsilon_{\max} = 1$. The saturation time was (ii) 0.11 s, (iii) 0.3 s for $\dot{\varepsilon} = 0.0109$ and (ii) 0.28 s, (iii) 1 s for $\dot{\varepsilon} = 0.0150$ ($n = 2$, $p = 10$ mm, $L_0 = 100$ mm). ...

We note that this is an example made by the small reduction in the initial chamber volume (since the initial chamber volume in ref [R2] is $\sim 3,200$ mm³), thus there are a lot of space for the actuator optimization.

Also, we have added more information in Supplementary Note 3.2 in page 5 and 6 of “Supplementary information” file as follows:

“3.2 Strain rate dependence

The output chamber pressure is a strain rate-dependent function due to the intrinsic limitation in the speed of air and the inflatable characteristics of the soft pneumatic actuators. As shown in Supplementary Fig. 6a, extending the STV at faster speeds could result in reduced output chamber pressures if the air from the inlet does not have enough time to move into the actuators to achieve the target chamber pressure state. Likewise, contracting the STV at higher speeds could result in increased output chamber pressures if the air from the actuator does not have enough time to exhaust. Also, the changes in air volume required for a soft actuator to reach a specific pressure is majorly dependent on the actuator’s initial chamber volume, geometry, and material properties. Since larger changes in air volume require longer time to reach the target pressure, the strain-rate dependency of the STV can be reduced, for example, by using an actuator with smaller initial chamber volume (Supplementary Fig. 6a (ii), (iii)).

In our system, the time required for the pressure to saturate for strain rate of 0.0109 s⁻¹ and 0.0150 s⁻¹ at $\varepsilon = \varepsilon_{\max}$ was 0.11 s and 0.28 s respectively for initial chamber volume = 27,615 mm³ and 0.3 s and 1 s respectively for initial chamber volume = 39,919 mm³.”

[R2] Mosadegh, B. et al. Pneumatic networks for soft robotics that actuate rapidly. *Adv. Funct. Mater.* **24**, 2163–2170 (2014).

Comment 4-2: Second, the valve suffers from continuous loss of the pressurised gas, which is not desirable for many applications as they require frequent replacement of the CO2 canister.

Response 4-2: The authors thank the reviewer’s comment.

The STV incurs a continuous loss of pressurized gas at intermediate output pressures. This is because intermediate output pressure states are achieved by appropriately controlling and maintaining the ratio of the inlet and outlet pneumatic resistance $R_{\text{in}}/R_{\text{out}}$ (between 0 and an extremely high value, in contrast to on/off valves that use only the two extreme values).

However, to achieve intermediate output pressures, previous research [R3] (which was previously the only research that studied on soft valves with proportional control capability to the best of our knowledge) has used a fixed outlet pneumatic resistance R_{out} which faces a trade-off between flow leakage and switching time. If R_{out} is too small, it can lead to a high loss of air and might result in pump systems failing to reach the desired pressure state (as the flowrate supplied by the pump must be significantly larger than the exhaust flowrate to achieve pressurized state). On the other hand, if R_{out} is too large, although the loss of the pressurized gas can be significantly reduced, it can significantly slow down the transition of output pressures from high to low at the same time. Therefore, R_{out} had to be carefully set within an acceptable range, which also causes a loss of compressed gas at all output states. The STV with a helical pinching mechanism resolve this previous issue by controlling both R_{in} and R_{out} in a programmable manner (when R_{in} increases, R_{out} decreases simultaneously and vice versa). This results in fast transitions of output from high to low without any flow leakage at high and low outputs (Supplementary Fig. 6 b(i)). Therefore, although STV do not completely terminate continuous loss of pressurized gas, we believe that STV provided an effective method to overcome previous challenges regarding the continuous loss of the pressurized gas. We have included this information in the Supplementary Table 1 in page 21 of “Supplementary information” file as follows:

Capability	Our valves	Micro-Fluidics ^{23, 40}	Bistable silicone valves ^{4, 49, 50, 53}	Digital, analog combinative valves ⁵²	Bidirectional check valves ⁴¹	Sheet-based valves ⁴⁷	Hysteretic valves ⁴⁶	Stretchable strain gauge ³⁶
Mechanism	Helical pinching	Linear pinching	Kinking	Kinking	Leaflet Flapping	Kinking	Slits in soft membrane	Channel elongation
Device shape	Tubular	Rectangular/Sheet	Cylindrical	Tubular	Tubular	Sheet	Tubular	Rectangular
Characteristic scale	Diameter = 5 mm	Thickness = 1 ~ 2.8 mm	Diameter = 27 mm, height = 34 mm	Diameter = 15 mm*	Channel length = 4 ~ 8 mm	Thickness = 2 mm**	Diameter = 5 mm	92 x 15 x 4 (mm)***
Practical flow rates for actuation	✓	—	✓	✓	✓	✓	✓	—
proportional control capabilities	✓	—	—	✓	—	—	—	—
Programmable proportional control	✓	—	—	▲	—	—	—	—
Operation with single, constant pressure source	✓	—	✓	✓	—	✓	✓	✓
Proportional self-sensing capabilities	✓	—	—	✓	—	—	—	✓
Inflow and outflow resistance control****	✓	—	—	—	—	—	—	—
Entirely soft	✓	✓	✓	—	✓	✓	✓	✓

* Estimated from Fig. 1c in⁵²

** Estimated from Fig. S13 in⁴⁷

*** Estimated from Fig. S13 in³⁶

▲ May be capable by changing the tension in the elastic band

Supplementary Table 1. Comparison of our valves to recently developed soft control devices.

**** The STV provide effective method to simultaneously modulate inflow and outflow resistance. Using a fixed outlet resistance R_{out} to achieve analog control faces a trade-off between flow leakage and switching time⁵². If R_{out} is too small, it can lead to a high loss of air and cause pump systems to fail to reach the desired pressure state. On the other hand, if R_{out} is too large, the transition of output pressures from high to low can be slowed down. Therefore, R_{out} had to be carefully set within an acceptable range, which also causes a loss of compressed gas at all output states. The STV with a helical pinching control both R_{in} and R_{out} in a programmable manner (when R_{in} increases, R_{out} decreases simultaneously and vice versa). This results in fast transitions of output from high to low with no flow leakage at high and low outputs.

From a practical point of view, the STV can last 64 minutes on average using a 95 g CO₂ canister and we believe this is not unusably short period of time, especially considering that portable demonstrations using soft valves previously had longevity of, for example, ~ 45 seconds using 57 g CO₂ canister and ~ 4 min using 306 g CO₂ canister [R4]. Also, since our 95 g CO₂ canister was not the heaviest type when considering for a canister, a frequent replacement

of the canister during practical use can be alleviated simply by using a larger CO₂ canister. For example, a rough calculation of using a 306 g CO₂ canister with our STV yields longevity of ~ 200 min, which we believe is not significantly short for many applications. Furthermore, we showed that the longevity can be further increased for more specific applications by adding an inlet resistor (Supplementary Note 3.3 and Supplementary Fig. 6b) which can approximately yield longevity of ~ 322 min using a 306 g CO₂ canister.

[R3] Decker, C. J. et al. Programmable soft valves for digital and analog control. *Proc. Natl. Acad. Sci. USA* **119**, <https://doi.org:10.1073/pnas.2205922119> (2022).

[R4] Drotman, D., Jadhav, S., Sharp, D., Chan, C. & Tolley, M. T. Electronics-free pneumatic circuits for controlling soft-legged robots. *Science Robotics* **6**, eaay2627 (2021).

Comment 4-3: Third, the authors argued that their proposed valve can be used for intelligent sensing, control, and perception. However, none of the application demonstrations focused on how the full cycle of sensing, control, and perception can be achieved.

Response 4-3: The authors thank the reviewer's comment.

In widely used pneumatic-based systems, many existing valves relied on electronic sensors and controllers. This not only sacrifice compactness and softness of the system but required a full cycle of (i) sensors reacting to a physical or environmental variables, (ii) sensors transducing the data into electrical signals, (iii) reading and processing the electrical data with a controller, and (iv) sending electrical signals to valves for action. However, The STV mechanically embeds the functions of soft stretchable sensors and electric valves into its fully soft, compact form factor, transducing self-sensed proportional strain directly into output pressures to realize electronics-free continuous control of soft robots. This is consistent with what we claimed in the manuscript: the STV “integrates the functional capabilities of sensors and control valves to directly transform applied tensile strain into distinctive steady-state output pressure states using only a single, constant pressure source”.

Therefore, since the mechanical sensing and control of the STV does not take place in multiple steps but rather in simultaneous manner, we could not focus on how the full cycle of sensing and control can be achieved, but instead focused on how the STVs can replace electronic components (controller, sensor, valve) and how they can be programmed and affect the output pressure curves.

Comment 4-4: Undoubtedly, the proposed design offers some benefits towards electronic-free robotic control, but at the same time, it brings some other complexities and challenges to a soft robotic system. Considering the above and based on the comments from other reviewers, the reviewer is not convinced that the current manuscript should be accepted for publication in Nature Communication.

Response 4-4: The authors thank the reviewer's comment.

As the reviewer pointed out, some complexities and challenges to a soft robotic system remain. However, we believe these challenges are not brought by the STV in exchange for its new capabilities, but rather have been as a fundamental trade-off when using soft pneumatic robotic system in general. For example, compare to rigid actuators, soft actuators have more safe and adaptable aspects, but face an expansion of the chamber with high volume change and therefore require significantly more air than a rigid actuator to achieve a target pressure. This challenge is not brought by continuously controlling an actuator but is fundamentally due to the inflations in soft actuators (please refer to Response 4-1). Instead, we believe that STV has resolved some of the existing challenges such as the previous trade-offs between flow leakage and switching speed (please refer to Response 4-2).

Inflatable soft robots that use intrinsically compliant components have great potential to provide adaptability to unstructured environments and safety while interacting with humans, contrary to their traditional rigid counterparts. However, their control systems highly relied on rigid solenoid valves and bulky electronic components, creating general and practical hurdles for applications in everyday life. Specifically, sensors, controllers, and regulators had to be connected via complex codes and wires. However, integrating rigid electronic components into soft robots undermines their primary benefit of mechanical compliance and adaptability, while tethering soft robots with externally located electronics limits their operating range. As a result, applications are greatly limited, including safe human-robot interactions (HRIs), or in environments where electronic devices are difficult to use, such as in vivo, underwater, or in the presence of spark ignitions. Regarding this context, the STV can provide strain-proportional control of soft actuators and replaces traditional electronics through the novel "helical pinching" mechanism, which enables the direct integration of sensing and control in a soft and compact form factor with mechanically programmable output pressure curves.

The comments that reviewer mentioned are very valuable. We appreciate the reviewer's input and acknowledgement of the importance of those applications. Although the issues that reviewer mentioned is not unique to STV (but rather a general challenge facing the field of soft robotics), we are aware that STV may be lacking in some practical point of view with some specific applications. In the 1st and 2nd revision process, we addressed these issues as best we could to clearly show the capability and limitations of the STV, which has significantly improved our manuscript. We hope the concerns of the reviewer has become resolved.

However, we must also admit that while there have been significant advancements in the design and capabilities of soft robots, there has always been a large trade-off between softness, compactness, intelligence, perception, complexities, etc. The STV overcame this trade-off, which is a significant breakthrough. Therefore, the STV is highly advantageous for use in fully soft systems, human-robot interactions, and possibly in complex soft robotic system with multiple independently controllable soft actuators. We believe our research on the STV represents a significant step forward in the field of soft robotics and has the potential to inspire further advances in this area of research, and therefore could be published in Nature Communications.

Comment 4-5: Another minor comment below:

1- We found the simulation to be in reasonable agreement with the analytical model shown in Supplementary Fig. 6 and the cross-sectional micro-CT images shown in Fig. 2b. Supplementary Fig. 6 seems to be displaced during the revision, please double check.

Response 4-5: The authors thank the reviewer's valuable comment. We checked and found that the reviewer was right. We have revised the "Pressure curve profile programming" section in page 7 of the manuscript as follows:

"We found the simulation to be in reasonable agreement with the analytical model shown in Supplementary Fig. 8"

REVIEWERS' COMMENTS

Reviewer #2 (Remarks to the Author):

I would like to thank the authors for once again making significant improvements to the manuscript. Overall I find the motivation of the work, the description and the figures far clearer. I have a couple of minor considerations for the authors:

Could some of the Supplementary figures/results be included in the main body of the text. There is capacity, and this would improve readability. For example some amount of S6-7.

Some of the figures in the Supplementary section seem quite low resolution/blurry, this could be improved.

Figure 3 – I feel the graphics could be improved – the black gripper on black background is hard to see.

Figure 4 c – typically you would expect to see multiple repeats of this motion.

Reviewer #3 (Remarks to the Author):

Third review of “A Soft, Self-Sensing Tensile Valve for Perceptive Soft Robots” (NCOMMS-22-37267B) by Jun Kyu Choe, Junsoo Kim, Hyeonseo Song, Joonbum Bae, and Jiyun Kim

This work describes the design and characterization of a soft pneumatic valve that produces an analog pressure output in response to tensile strain. We thank the authors for revising the relevant entries in supplementary table 1 in response to our previous comment. As stated in our earlier review, we believe that the soft and compact pneumatic strain sensor introduced in this work represents an interesting and timely contribution to the ongoing effort in soft robotics toward developing soft sensors and transducers that directly output fluidic pressure signals. Furthermore, the authors' detailed characterization of their device, included during the two rounds of revision, significantly adds to the comprehensiveness of the work and provides key information necessary to enable future customization and implementation of their device by other researchers. As the current version of the manuscript satisfactorily addresses our earlier questions and concerns, we are happy to recommend its publication.

Reviewer #4 (Remarks to the Author):

The authors have made reasonable changes in the manuscript. The reviewer is now happy to recommend publication of the revised manuscript.

3rd Response to reviewers' comments for the manuscript – “A Soft, Self-Sensing Tensile Valve for Perceptive Soft Robots”

Response to reviewer #2's comments

General Comment: I would like to thank the authors for once again making significant improvements to the manuscript. Overall I find the motivation of the work, the description and the figures far clearer.

Response for General Comment: We are grateful for the reviewer's positive feedback regarding the modifications we have made to our manuscript. We are pleased to hear that these revisions have contributed to the overall improvement of the manuscript. We sincerely appreciate the time and effort the reviewer dedicated in reviewing our work.

Comment 2-1: I have a couple of minor considerations for the authors: Could some of the Supplementary figures/results be included in the main body of the text. There is capacity, and this would improve readability. For example some amount of S6-7.

Response 2-1: The authors thank the reviewer's valuable comment. We have added some of the Supplementary Fig. 6 and Supplementary Note 3 information in “Untethered and electronics-free soft gripper” section in page 9 of the manuscript as follows:

“From a practical point of view, the targeting of the output chamber pressure was immediately set according to the strain of the STV, but for soft actuators, the achievement of the target pressure was strain rate-dependent that highly depended on the soft actuator characteristics such as the initial chamber volume (Supplementary Fig. 6a). The longevity of the CO₂ canister (95 g) using the STV was 64 minutes on average but could be increased to 100 minutes using an additional inlet resistor (Supplementary Fig. 6b). The STV showed good reproducibility with similar output from 5 different samples (Supplementary Fig. 6c), durability up to 10,000 full cycles, and could be simply repaired after sharp cuts or detachment of the outer tube (Supplementary Fig. 6d). See Supplementary Note 3 for more information related to the STV during practical use.”

Also We have added more information related to Supplementary Fig. 7 in “Autonomous and self-adaptive soft exosuit” section in page 10 of the manuscript as follows:

“The STV with a soft, compact, and linear form can also provide conformable and safe human–machine interactions (see Supplementary Fig. 7 for STV behavior under bending, buckling, kinking, and compression).”

Comment 2-2: Some of the figures in the Supplementary section seem quite low resolution/blurry, this could be improved.

Figure 3 – I feel the graphics could be improved – the black gripper on black background is hard to see.

Figure 4 c – typically you would expect to see multiple repeats of this motion.

Response 2-2: The authors thank the reviewer’s valuable comments. In response to the reviewer’s comments, we have amended some of the figures in the supplementary section by adjusting sharpness, size, placement, brightness and contrast of the contents. We modified Supplementary Fig. 6e, 7, and 12b for graphical improvement as follows:

Supplementary Fig. 6. Considerable factors of the STV during practical use. ...

Supplementary Fig. 7. STV under different mechanical loads. ...

Supplementary Fig. 12. Chamber pressure curve profile analysis setup. ...

Also, while the black gripper body on black background in Fig. 3 is hard to see, we found that changing the background into white color can cause similar problem as the soft actuators are white. Therefore, we similarly adjusted sharpness, brightness, and contrast of the contents in Fig. 3 for graphical improvement as follows:

Fig. 3. Untethered and electronics-free soft gripper. ...

Also, regarding Fig. 4c:

The STV is the main focus of our study, and we used the exosuit application as one of the examples to illustrate how the STV can be effectively exploited. Specifically, Fig. 4c focus on how the STV can be exploited to sense and transfer arm flexion angles to continuous, increasing output pressure states, which result in reduced gravitational torque compared to undressed condition without using STV or static pressure conditions.

The STV we used ($n = 2$) can yield a minor hysteresis during cyclic motion, and this hysteresis can be further reduced or magnified by the STV parameters, arm flexion speed, and various characteristics of the soft actuator. Since there are a lot of variables that can complicate the understanding of the message we intended to deliver in Fig. 4c, and there are already an extensive study that focus on the loading/unloading behavior, repeatability, durability, etc. of the STV (Supplementary Fig. 4, 6), we carefully decided to only provide the continuous and increasing function in Fig. 4c, which STVs can provide in general.

Response to reviewer #3's comments

General Comment: Third review of "A Soft, Self-Sensing Tensile Valve for Perceptive Soft Robots" (NCOMMS-22-37267B) by Jun Kyu Choe, Junsoo Kim, Hyeonseo Song, Joonbum Bae, and Jiyun Kim

This work describes the design and characterization of a soft pneumatic valve that produces an analog pressure output in response to tensile strain. We thank the authors for revising the relevant entries in supplementary table 1 in response to our previous comment. As stated in our earlier review, we believe that the soft and compact pneumatic strain sensor introduced in this work represents an interesting and timely contribution to the ongoing effort in soft robotics toward developing soft sensors and transducers that directly output fluidic pressure signals. Furthermore, the authors' detailed characterization of their device, included during the two rounds of revision, significantly adds to the comprehensiveness of the work and provides key information necessary to enable future customization and implementation of their device by other researchers. As the current version of the manuscript satisfactorily addresses our earlier questions and concerns, we are happy to recommend its publication.

Response for General Comment: We are delighted to hear that you recommend the publication of our manuscript. The reviewer's valuable comments and suggestions have played a crucial role in shaping and improving our work. We extend our heartfelt gratitude for the time and effort the reviewer devoted to reviewing our manuscript.

Response to reviewer #4's comments

General Comment: The authors have made reasonable changes in the manuscript. The reviewer is now happy to recommend publication of the revised manuscript.

Response for General Comment: We deeply appreciate the reviewer's positive feedback regarding the revisions we made to our manuscript. We're glad to know that these changes have strengthened the manuscript. Thank you again for your valuable contribution.